# Trust Region-Guided Proximal Policy Optimization

Yuhui Wang , Hao He , Xiaoyang Tan , Yaozhong Gan

College of Computer Science and Technology,Nanjing University of Aeronautics and Astronautics
MIIT Key Laboratory of Pattern Analysis and Machine Intelligence
Collaborative Innovation Center of Novel Software Technology and Industrialization
*{y.wang, hugo, x.tan, yzgancn}@nuaa.edu.cn*

## Abstract

Proximal policy optimization (PPO) is one of the most popular deep reinforcement learning (RL) methods, achieving state-of-the-art performance across a wide range of challenging tasks. However, as a model-free RL method, the success of PPO relies heavily on the effectiveness of its exploratory policy search. In this paper, we give an in-depth analysis on the exploration behavior of PPO, and show that PPO is prone to suffer from the risk of lack of exploration especially under the case of bad initialization, which may lead to the failure of training or being trapped in bad local optima. To address these issues, we proposed a novel policy optimization method, named Trust Region-Guided PPO (TRGPPO), which adaptively adjusts the clipping range within the trust region. We formally show that this method not only improves the exploration ability within the trust region but enjoys a better performance bound compared to the original PPO as well. Extensive experiments verify the advantage of the proposed method.

## 1 Introduction

Deep model-free reinforcement learning has achieved great successes in recent years, notably in video games [11], board games [19], robotics [10], and challenging control tasks [17, 5]. Among others, policy gradient (PG) methods are commonly used model-free policy search algorithms [14]. However, the first-order optimizer is not very accurate for curved areas. One can get overconfidence and make bad moves that ruin the progress of the training. Trust region policy optimization (TRPO) [16] and proximal policy optimization (PPO) [18] are two representative methods to address this issue. To ensure stable learning, both methods impose a constraint on the difference between the new policy and the old one, but with different policy metrics.

In particular, TRPO uses a divergence between the policy distributions (total variation divergence or KL divergence), whereas PPO uses a probability ratio between the two policies[1]. The divergence metric is proven to be theoretically-justified as optimizing the policy within the divergence constraint (named trust region) leads to guaranteed monotonic performance improvement. Nevertheless, the complicated second-order optimization involved in TRPO makes it computationally inefficient and difficult to scale up for large scale problems. PPO significantly reduces the complexity by adopting a clipping mechanism which allows it to use a first-order optimization. PPO is proven to be very effective in dealing with a wide range of challenging tasks while being simple to implement and tune.

However, how the underlying metric adopted for policy constraints influence the behavior of the algorithm is not well understood. It is normal to expect that the different metrics will yield RL algorithms with different exploration behaviors. In this paper, we give an in-depth analysis on the

exploration behavior of PPO, and show that the ratio-based metric of PPO tends to continuously weaken the likelihood of choosing an action in the future if that action is not preferred by the current policy. As a result, PPO is prone to suffer from the risk of lack of exploration especially under the case of bad initialization, which may lead to the failure of training or being trapped in bad local optima.

To address these issues, we propose an enhanced PPO method, named Trust Region-Guided PPO (TRGPPO), which is theoretically justified by the improved exploration ability and better performance bound compared to the original PPO. In particular, TRGPPO constructs a connection between the ratio-based metric and trust region-based one, such that the resulted ratio clipping mechanism allows the constraints imposed on the less preferred actions to be relaxed. This effectively encourages the policy to explore more on the potential valuable actions, no matter whether they were preferred by the previous policies or not. Meanwhile, the ranges of the new ratio-based constraints are kept within the trust region; thus it would not harm the stability of learning. Extensive results on several benchmark tasks show that the proposed method significantly improves both the policy performance and the sample efficiency. Source code is available at https://github.com/wangyuhuix/TRGPPO.

## 2 Related Work

Many researchers have tried to improve proximal policy learning from different perspectives. Chen et al. also presented a so-called "adaptive clipping mechanism" for PPO [3]. Their method adaptively adjusts the scale of policy gradient according to the significance of state-action. They did *not* make any alteration on the clipping mechanism of PPO, while our method adopts a newly adaptive clipping mechanism. Fakoor et al. used proximal learning with penalty on KL divergence to utilize the off-policy data, which could effectively reduce the sample complexity [6]. In our previous work, we also introduced trust region-based clipping to improve boundness on policy of PPO [22]. While in this work, we use the trust region-based criterion to guide the clipping range adjustment, which requires additional computation but is more flexible and interpretable.

Several methods have been proposed to improve exploration in recent research. Osband et al. tried to conduct consistent exploration using posterior sampling method [12]. Fortunato et al. presented a method named NoisyNet to improve exploration by generating perturbations of the network weights [7]. Another popular algorithm is the soft actor-critic method (SAC) [9], which maximizes expected reward and entropy simultaneously.

## 3 Preliminaries

A Markov Decision Processes (MDP) is described by the tuple $(\mathcal{S}, \mathcal{A}, \mathcal{T}, c, \rho_1, \gamma)$. $\mathcal{S}$ and $\mathcal{A}$ are the state space and action space; $\mathcal{T} : \mathcal{S} \times \mathcal{A} \times \mathcal{S} \to \mathbb{R}$ is the transition probability distribution; $c : \mathcal{S} \times \mathcal{A} \to \mathbb{R}$ is the reward function; $\rho_1$ is the distribution of the initial state $s_1$, and $\gamma \in (0, 1)$ is the discount factor. The return is the accumulated discounted reward from timestep $t$ onwards, $R_t^\gamma = \sum_{k=0}^\infty \gamma^k c(s_{t+k}, a_{t+k})$. The performance of a policy $\pi$ is defined as $\eta(\pi) = \mathbb{E}_{s \sim \rho^\pi, a \sim \pi}[c(s, a)]$ where $\rho^\pi(s) = (1 - \gamma) \sum_{t=1}^\infty \gamma^{t-1} \rho_t^\pi(s)$, $\rho_t^\pi$ is the density function of state at time $t$. Policy gradients methods [20] update the policy by the following surrogate performance objective, $L_{\pi_{\text{old}}}(\pi) = \mathbb{E}_{s \sim \rho^{\pi_{\text{old}}}, a \sim \pi_{\text{old}}}\left[\frac{\pi(a|s)}{\pi_{\text{old}}(a|s)} A^{\pi_{\text{old}}}(s, a)\right] + \eta(\pi_{\text{old}})$, where $\pi(a|s)/\pi_{\text{old}}(a|s)$ is the *probability ratio* between the new policy $\pi$ and the old policy $\pi_{\text{old}}$, $A^\pi(s, a) = \mathbb{E}[R_t^\gamma | s_t = s, a_t = a; \pi] - \mathbb{E}[R_t^\gamma | s_t = s; \pi]$ is the advantage value function of policy $\pi$. Let $D_{\text{KL}}^{\text{s}}(\pi_{\text{old}}, \pi) \triangleq D_{\text{KL}}(\pi_{\text{old}}(\cdot|s) || \pi(\cdot|s))$, Schulman et al. [16] derived the following performance bound:

**Theorem 1.** *Define that* $C = \max_{s,a} |A^{\pi_{\text{old}}}(s, a)| 4\gamma / (1 - \gamma)^2$, $M_{\pi_{\text{old}}}(\pi) = L_{\pi_{\text{old}}}(\pi) - C \max_{s \in \mathcal{S}} D_{\text{KL}}^{\text{s}}(\pi_{\text{old}}, \pi)$. *We have* $\eta(\pi) \geq M_{\pi_{\text{old}}}(\pi), \eta(\pi_{\text{old}}) = M_{\pi_{\text{old}}}(\pi_{\text{old}})$.

This theorem implies that maximizing $M_{\pi_{\text{old}}}(\pi)$ guarantee non-decreasing of the performance of the new policy $\pi$. To take larger steps in a robust way, TRPO optimizes $L_{\pi_{\text{old}}}(\pi)$ with the constraint $\max_{s \in \mathcal{S}} D_{\text{KL}}^{\text{s}}(\pi_{\text{old}}, \pi) \leq \delta$, which is called the *trust region*.

## 4 The Exploration Behavior of PPO

In this section will first give a brief review of PPO and then show that how PPO suffers from an exploration issue when the initial policy is sufficiently far from the optimal one.

PPO imposes the policy constraint through a clipped surrogate objective function:

$$L_{\pi_{\text{old}}}^{\text{CLIP}}(\pi) = \mathbb{E}\left[\min\left(\frac{\pi(a|s)}{\pi_{\text{old}}(a|s)}A^{\pi_{\text{old}}}(s,a), clip\left(\frac{\pi(a|s)}{\pi_{\text{old}}(a|s)}, l_{s,a}, u_{s,a}\right)A^{\pi_{\text{old}}}(s,a)\right)\right] \quad (1)$$

where $l_{s,a} \in (0,1)$ and $u_{s,a} \in (1,+\infty)$ are called the lower and upper *clipping range* on state-action $(s,a)$. The probability ratio $\pi(a|s)/\pi_{\text{old}}(a|s)$ will be clipped once it is out of $(l_{s,a}, u_{s,a})$. Therefore, such clipping mechanism could be considered as a constraint on policy with ratio-based metric, i.e., $l_{s,a} \leq \pi(a|s)/\pi_{\text{old}}(a|s) \leq u_{s,a}$, which can be rewritten as, $-\pi_{\text{old}}(a|s)(1 - l_{s,a}) \leq \pi(a|s) - \pi_{\text{old}}(a|s) \leq \pi_{\text{old}}(a|s)(u_{s,a} - 1)$. We call $(\mathcal{L}_{\pi_{\text{old}}}^{l}(s,a), \mathcal{U}_{\pi_{\text{old}}}^{u}(s,a)) \triangleq (-\pi_{\text{old}}(a|s)(1 - l_{s,a}), \pi_{\text{old}}(a|s)(u_{s,a} - 1))$ the *feasible variation range* of policy $\pi$ w.r.t. $\pi_{\text{old}}$ on state-action $(s,a)$ with the clipping range setting $(l, u)$, which is a measurement on the allowable change of policy $\pi$ on state-action $(s,a)$.

Note that the original PPO adopts a constant setting of clipping range, i.e., $l_{s,a} = 1 - \epsilon, u_{s,a} = 1 + \epsilon$ for any $(s,a)$ [18]. The corresponding feasible variation range is $(\mathcal{L}_{\pi_{\text{old}}}^{1-\epsilon}(s,a), \mathcal{U}_{\pi_{\text{old}}}^{1+\epsilon}(s,a)) = (-\pi_{\text{old}}(a|s)\epsilon, \pi_{\text{old}}(a|s)\epsilon)$. As can be seen, given an optimal action $a_{\text{opt}}$ and a sub-optimal one $a_{\text{subopt}}$ on state $s$, if $\pi_{\text{old}}(a_{\text{opt}}|s) < \pi_{\text{old}}(a_{\text{subopt}}|s)$, then $|(\mathcal{L}_{\pi_{\text{old}}}^{1-\epsilon}(s, a_{\text{opt}}), \mathcal{U}_{\pi_{\text{old}}}^{1+\epsilon}(s, a_{\text{opt}}))| < |(\mathcal{L}_{\pi_{\text{old}}}^{1-\epsilon}(s, a_{\text{subopt}}), \mathcal{U}_{\pi_{\text{old}}}^{1+\epsilon}(s, a_{\text{subopt}}))|$. This means that the allowable change of the likelihood on optimal action, i.e., $\pi(a_{\text{opt}}|s)$, is smaller than that of $\pi(a_{\text{subopt}}|s)$. Note that $\pi(a_{\text{opt}}|s)$ and $\pi(a_{\text{subopt}}|s)$ are in a zero-sum competition, such unequal restriction may continuously weaken the likelihood of the optimal action and make the policy trapped in local optima. We now give a formal illustration.

---

**Algorithm 1** Simplified Policy Iteration with PPO

---

1: Initialize a policy $\pi_0$, $t \leftarrow 0$.
2: **repeat**
3:    Sample an action $\hat{a}_t \sim \pi_t$.
4:    Get the new policy $\pi_{t+1}$ by optimizing the empirical surrogate objective function of PPO based on $\hat{a}_t$:

$$\hat{\pi}_{t+1}(a) = \begin{cases} \pi_t(a)u_a & a = \hat{a}_t \text{ and } c(a) > 0 \\ \pi_t(a)l_a & a = \hat{a}_t \text{ and } c(a) < 0 \\ \pi_t(a) - \frac{\pi_t(\hat{a}_t)u_{\hat{a}_t} - \pi_t(\hat{a}_t)}{|\mathcal{A}| - 1} & a \neq \hat{a}_t \text{ and } c(\hat{a}_t) > 0 \\ \pi_t(a) + \frac{\pi_t(\hat{a}_t)(1 - l_{\hat{a}_t})}{|\mathcal{A}| - 1} & a \neq \hat{a}_t \text{ and } c(\hat{a}_t) < 0 \\ \pi_t(a) & c(\hat{a}_t) = 0 \end{cases} \quad (2)$$

5:    $\pi_{t+1} = Normalize(\hat{\pi}_{t+1})^2$. $t \leftarrow t + 1$.
6: **until** $\pi_t$ converge

---

We investigate the exploration behavior of PPO under the discrete-armed bandit problem, where there are no state transitions and the action space is discrete. The objective function of PPO in this problem is $L_{\pi_{\text{old}}}^{\text{CLIP}}(\pi) = \mathbb{E}\left[\min\left(\frac{\pi(a)}{\pi_{\text{old}}(a)}c(a), clip\left(\frac{\pi(a)}{\pi_{\text{old}}(a)}, l_a, u_a\right)c(a)\right)\right]$. *Let* $\mathcal{A}^+ \triangleq \{a \in \mathcal{A}|c(a) > 0\}$, $\mathcal{A}^- \triangleq \{a \in \mathcal{A}|c(a) < 0\}$ *denote the actions which have positive and negative reward respectively, and* $\mathcal{A}_{\text{subopt}} = \mathcal{A}^+/\{a_{\text{opt}}\}$ *denote the set of the sub-optimal actions. Let* $a_{\text{opt}} = argmax_a c(a)$ *and* $a_{\text{subopt}} \in \mathcal{A}_{\text{subopt}}$ *denote the optimal [3] and a sub-optimal action.* Let us consider a simplified online policy iteration algorithm with PPO. As presented in Algorithm 1, the algorithm iteratively sample an action $\hat{a}_t$ based on the old policy $\pi_{\text{old}}$ at each step and obtains a new policy $\pi_{\text{new}}$.

We measure the exploration ability by the expected distance between the learned policy $\pi_t$ and the optimal policy $\pi^*$ after $t$-step learning, i.e., $\Delta_{\pi_0, t} \triangleq \mathbb{E}_{\pi_t}[\|\pi_t - \pi^*\|_\infty|\pi_0]$, where $\pi^*(a_{\text{opt}}) = 1$, $\pi^*(a) = 0$ for $a \neq a_{\text{opt}}$, $\pi_0$ is the initial policy, $\pi_t$ is a stochastic element in the policy space and

depends on the previous sampled actions $\{a_{t'}\}_{t'=1}^{t-1}$ (see eq. (2)). *Note that smaller $\Delta_{\pi_0, t}$ means better exploration ability, as it is closer to the optimal policy.* We now derive the exact form of $\Delta_{\pi_0, t}$.

**Lemma 1.** $\Delta_{\pi_0, t} \triangleq \mathbb{E}_{\pi_t} [\|\pi_t - \pi^*\|_\infty | \pi_0] = 1 - \mathbb{E}_{\pi_t} [\pi_t(a_{\mathrm{opt}}) | \pi_0]$.

**Lemma 2.** $\mathbb{E}_{\pi_{t+1}} [\pi_{t+1}(a) | \pi_0] = \mathbb{E}_{\pi_t} [\mathbb{E}_{\pi_{t+1}} [\pi_{t+1}(a) | \pi_t] | \pi_0]$.

We provide all the proofs in Appendix E. Lemma 1 implies that we can obtain the exploration ability $\Delta_{\pi_0, t}$ by computing the expected likelihood of the optimal action $a_{\mathrm{opt}}$, i.e., $\mathbb{E}_{\pi_t} [\pi_t(a_{\mathrm{opt}}) | \pi_0]$. And Lemma 2 shows an iterative way to compute the exploration ability. By eq. (2), for action $a$ which satisfies $c(a) > 0$, we have

$$\mathbb{E}_{\pi_{t+1}} [\pi_{t+1}(a) | \pi_t] = \pi_t(a) + \left[ \pi_t^2(a)(u_a - 1) - \sum_{a^+ \in \mathcal{A}^+ / \{a\}} \frac{\pi_t^2(a^+)}{|\mathcal{A}| - 1}(u_{a^+} - 1) + \sum_{a^- \in \mathcal{A}^-} \frac{\pi_t^2(a^-)}{|\mathcal{A}| - 1}(1 - l_{a^-}) \right] \quad (3)$$

This equation provides a explicit form of the case when the likelihood of action $a$ would decrease. That is, if the second term in RHS of eq. (3) is negative, then the likelihood on action $a$ would decrease. This means that the initialization of policy $\pi_0$ profoundly affects the future policy $\pi_t$. Now we show that if the policy $\pi_0$ initializes from a bad one, $\pi(a_{\mathrm{opt}})$ may continuously be decreased. Formally, for PPO, we have the following theorem:

**Theorem 2.** *Given initial policy $\pi_0$, if $\pi_0^2(a_{\mathrm{opt}}) \cdot |\mathcal{A}| < \sum_{a_{\mathrm{subopt}} \in \mathcal{A}_{\mathrm{subopt}}} \pi_0^2(a_{\mathrm{subopt}}) - \sum_{a^- \in \mathcal{A}^-} \pi_0^2(a^-)$, then we have*

*(i)* $\sum_{a_{\mathrm{subopt}} \in \mathcal{A}_{\mathrm{subopt}}} \pi_0(a_{\mathrm{subopt}}) < \sum_{a_{\mathrm{subopt}} \in \mathcal{A}_{\mathrm{subopt}}} \mathbb{E}_{\pi_1^{\mathrm{PPO}}} [\pi_1^{\mathrm{PPO}}(a_{\mathrm{subopt}}) | \pi_0] < \cdots < \sum_{a_{\mathrm{subopt}} \in \mathcal{A}_{\mathrm{subopt}}} \mathbb{E}_{\pi_t^{\mathrm{PPO}}} [\pi_t^{\mathrm{PPO}}(a_{\mathrm{subopt}}) | \pi_0]$;

*(ii)* $\pi_0(a_{\mathrm{opt}}) > \mathbb{E}_{\pi_1^{\mathrm{PPO}}} [\pi_1^{\mathrm{PPO}}(a_{\mathrm{opt}}) | \pi_0] > \cdots > \mathbb{E}_{\pi_t^{\mathrm{PPO}}} [\pi_t^{\mathrm{PPO}}(a_{\mathrm{opt}}) | \pi_0]$;

*(iii)* $\Delta_{\pi_0, 0} < \Delta_{\pi_0, 1}^{\mathrm{PPO}} < \cdots < \Delta_{\pi_0, t}^{\mathrm{PPO}}$.

Conclusion (i) and (ii) implies that if the optimal action $a_{\mathrm{opt}}$ is relatively less preferred than the sub-optimal action $a_{\mathrm{subopt}}$ by the initial policy, then the preference of choosing the optimal action would continue decreasing while that of the sub-optimal action would continue increasing. This is because the feasible variation of probability on the optimal action $\pi(a_{\mathrm{opt}})$ is larger than that on the sub-optimal one $\pi(a_{\mathrm{subopt}})$, increasing probability on the latter one could diminish the former one. Conclusion (iii) implies that the policy of PPO is expected to diverge from the optimal one (in terms of the infinity metric). We give a simple example below.

**Example 1.** *Consider a three-armed bandit problem, the reward function is $c(a_{\mathrm{opt}}) = 1, c(a_{\mathrm{subopt}}) = 0.5, c(a_{\mathrm{worst}}) = -50$. The initial policy is $\pi_0(a_{\mathrm{opt}}) = 0.2, \pi_0(a_{\mathrm{subopt}}) = 0.6, \pi_0(a_{\mathrm{worst}}) = 0.2$. The hyperparameter of PPO is $\epsilon = 0.2$. We have $\Delta_{\pi_0, 0}^{\mathrm{PPO}} = 0.8, \Delta_{\pi_0, 1}^{\mathrm{PPO}} = 0.824, \ldots, \Delta_{\pi_0, 6}^{\mathrm{PPO}} \approx 0.999$, which means the policy diverges from the optimal one.*

Note that the case that the optimal action $a_{\mathrm{opt}}$ is relatively less preferred by the initial policy may be avoided in discrete action space, where we can use uniform distribution as initial policy. However, such a case could hardly be avoided in the high dimensional action space, where the policy is possibly initialized far from the optimal one. We have experimented Example 1 and a continuous-armed bandit problem with random initialization for multiple trials; about 30% of the trials were trapped in the local optima. See Section 6.1 for more detail.

In summary, PPO with constant clipping range could lead to an exploration issue when the policy is initialized from a bad one. However, eq. (3) inspires us a method to address this issue – enlarging the clipping range $(l_a, u_a)$ when the probability of the old policy $\pi_{\mathrm{old}}(a)$ is small.

## 5 Method

### 5.1 Trust Region-Guided PPO

In the previous section, we have concluded that the constant clipping range of PPO could lead to an exploration issue. We consider how to adaptively adjust the clipping range to improve the exploration

behavior of PPO. The new clipping range $(l^\delta_{s,a}, u^\delta_{s,a})$, where $\delta$ is a hyperparameter, is set as follows:

$$l^\delta_{s,a} = \min_\pi \left\{ \frac{\pi(a|s)}{\pi_{\text{old}}(a|s)} : D^s_{\text{KL}}(\pi_{\text{old}}, \pi) \leq \delta \right\}, u^\delta_{s,a} = \max_\pi \left\{ \frac{\pi(a|s)}{\pi_{\text{old}}(a|s)} : D^s_{\text{KL}}(\pi_{\text{old}}, \pi) \leq \delta \right\} \quad (4)$$

To ensure the new adaptive clipping range would not be over-strict, an additional truncation operation is attached: $l^{\delta,\epsilon}_{s,a} = \min(l^\delta_{s,a}, 1-\epsilon), u^{\delta,\epsilon}_{s,a} = \max(u^\delta_{s,a}, 1+\epsilon)$. This setting of clipping range setting could be motivated from the following perspectives.

First, the clipping range is related to the policy metric of constraint. Both TRPO and PPO imposes a constraint on the difference between the new policy and the old one. TRPO uses the divergence metric of the distribution, i.e., $D^s_{\text{KL}}(\pi_{\text{old}}, \pi) = \mathbb{E}_a \left[ \log \frac{\pi_{\text{old}}(a|s)}{\pi(a|s)} \right] \leq \delta$ for all $s \in \mathcal{S}$, which is more theoretically-justified according to Theorem 1. Whereas PPO uses a ratio-based metric on each action, i.e., $1 - \epsilon \leq \frac{\pi(a|s)}{\pi_{\text{old}}(a|s)} \leq 1 + \epsilon$ for all $a \in \mathcal{A}$ and $s \in \mathcal{S}$. The divergence-based metric is averaged over the action space while the ratio-based one is an element-wise one on each action point. If the policy is restricted within a region with the ratio-based metric, then it is also constrained within a region with divergence-based one, but not vice versa. Thus the probability ratio-based metric constraint is somewhat more strict than the divergence-based one. Our method connects these two underlying metrics − adopts the probability ratio-based constraint while getting closer to the divergence metric.

Second, a different underlying metric of the policy difference may result in different algorithm behavior. In the previous section, we have concluded that PPO's metric with constant clipping range could lead to an exploration issue, due to that it imposes a relatively strict constraint on actions which are not preferred by the old policy. Therefore, we wish to relax such constraint by enlarging the upper clipping range while reducing the lower clipping range. Fig. 1a shows the clipping range of TRGPPO and PPO. For TRGPPO (blue curve), as $\pi_{\text{old}}(a|s)$ gets smaller, the upper clipping range increases while the lower one decreases, which means the constraint is relatively relaxed as $\pi_{\text{old}}(a|s)$ gets smaller. This mechanism could encourage the agent to explore more on the potential valuable actions which are not preferred by the old policy. We will theoretically show that the exploration behavior with this new clipping range is better than that of with the constant one in Section 5.2.

Last but not least, although the clipping ranges are enlarged, it will not harm the stability of learning, as the ranges are kept within the trust region. We will show that this new setting of clipping range would not enlarge the policy divergence and has better performance bound compared to PPO in Section 5.3.

Our TRGPPO adopts the same algorithm procedure as PPO, except that it needs an additional computation of adaptive clipping range. We now present methods on how to compute the adaptive clipping range defined in (4) efficiently. For discrete action space, by using the KKT conditions, the problem (4) is transformed into solving the following equation w.r.t $X$.

$$g(\pi_{\text{old}}(a|s), X) \triangleq (1 - \pi_{\text{old}}(a|s)) \log \frac{1 - \pi_{\text{old}}(a|s)}{1 - \pi_{\text{old}}(a|s)X} - \pi_{\text{old}}(a|s) \log X = \delta \quad (5)$$

which has two solutions, one is for $l^\delta_{s,a}$ which is within $(0, 1)$, and another one is for $u^\delta_{s,a}$ which is within $(1, +\infty)$. We use MINPACK's HYBRD and HYBRJ routines [15] as the solver. To accelerate this computation procedure, we adopt two additional measures. First, we train a Deep Neural Network

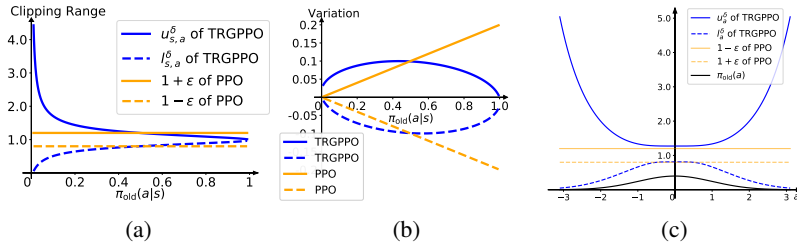

Figure 1: (a) and (b) plot the clipping range and the feasible variation range under different $\pi_{\text{old}}(a|s)$ for discrete action space task. (c) plots the clipping range under different $a$ for continuous action space task; the black curve plots the density of $\pi_{\text{old}}(a|s) = \mathcal{N}(a|0, 1)$.

(DNN) which input $\pi_{\text{old}}(a|s)$ and $\delta$, and approximately output the initial solution. Note that the solution in (5) only depends on the probability $\pi_{\text{old}}(a|s)$ and the hyperparameter $\delta$, and it is not affected by the dimension of the action space. Thus it is possible to train one DNN for all discrete action space tasks in advance. Second, with fixed $\delta$, we discretize the probability space and save all the solutions in advance. This clipping range computation procedure with these two acceleration measures only requires only additional 4% wallclock computation time of the original policy learning. See Appendix B.3 for more detail.

While for the continuous actions space task, we make several transformations to make the problem independent of the dimension of the action space, which makes it tractable to apply the two acceleration measures above. See Appendix B.2 for more detail.

## 5.2 Exploration Behavior

In this section, we will first give the property of the clipping range of TRGPPO, which could affect the exploration behavior (as discussed in Section 4). Then a comparison between TRGPPO and PPO on the exploration behavior will be provided.

**Lemma 3.** *For TRGPPO with hyperparameter $\delta$, we have $\frac{du_{s,a}^\delta}{d\pi_{\text{old}}(a|s)} < 0$, $\frac{dl_{s,a}^\delta}{d\pi_{\text{old}}(a|s)} > 0$.*

This result implies that the upper clipping range becomes larger as the preference on the action by the old policy $\pi_{\text{old}}(a|s)$ approaches zero, while the lower clipping range is on the contrary. This means that the constraints are relaxed on the actions which are not preferred by the old policy, such that it would encourage the policy to explore more on the potential valuable actions, no matter whether they were preferred by the previous policies or not.

We now give a formal comparison on the exploration behavior. As mentioned in Section 4, we measure the exploration ability by the expected distance between the learned policy $\pi_t$ and the optimal policy $\pi^*$ after $t$-step learning, i.e., $\Delta_{\pi_0,t} \triangleq \mathbb{E}_{\pi_t}\left[\|\pi_t - \pi^*\|_\infty | \pi_0\right]$. Smaller $\Delta_{\pi_0,t}$ means the better exploration ability. The exploration ability of TRGPPO is denoted as $\Delta_{\pi_0,t}^{\text{TRGPPO}}$ while that of PPO is denoted as $\Delta_{\pi_0,t}^{\text{PPO}}$. By eq. (3) and Lemma 3, we get the following conclusion.

**Theorem 3.** *For TRGPPO with hyperparameter $(\delta, \epsilon)$ and PPO with same $\epsilon$. If $\delta \leq g(\max_{a \in \mathcal{A}_{\text{subopt}}} \pi_t(a), 1 + \epsilon)$ for all $t$, then we have $\Delta_{\pi_0,t}^{\text{TRGPPO}} \leq \Delta_{\pi_0,t}^{\text{PPO}}$ for any $t$.*

This theorem implies that our TRGPPO has better exploration ability than PPO, with proper setting of the hyperparameter $\delta$.

## 5.3 Policy Divergence and Lower Performance Bound

To investigate how TRGPPO and PPO perform in practical, let us consider an empirical version of lower performance bound: $\hat{M}_{\pi_{\text{old}}}(\pi) = \hat{L}_{\pi_{\text{old}}}(\pi) - C\max_t D_{\text{KL}}^{s_t}(\pi_{\text{old}}, \pi)$, where $\hat{L}_{\pi_{\text{old}}}(\pi) = \frac{1}{T}\sum_{t=1}^{T}\left[\frac{\pi(a_t|s_t)}{\pi_{\text{old}}(a_t|s_t)}A_t\right] + \hat{\eta}^{\pi_{\text{old}}}$, $s_t \sim \rho_{\pi_{\text{old}}}, a_t \sim \pi_{\text{old}}(\cdot|s_t)$ are the sampled states and actions, where we assume $s_i \neq s_j$ for any $i \neq j$, $A_t$ is the estimated value of $A^{\pi_{\text{old}}}(s_t, a_t)$, $\hat{\eta}^{\pi_{\text{old}}}$ is the estimated performance of old policy $\pi_{\text{old}}$.

Let $\Pi_{\text{new}}^{\text{PPO}}$ denote the set of all the optimal solutions of the empirical surrogate objective function of PPO, and let $\pi_{\text{new}}^{\text{PPO}} \in \Pi_{\text{new}}^{\text{PPO}}$ denote the optimal solution which achieve minimum KL divergence over all optimal solutions, i.e., $D_{\text{KL}}^{s_t}(\pi_{\text{old}}, \pi_{\text{new}}^{\text{PPO}}) \leq D_{\text{KL}}^{s_t}(\pi_{\text{old}}, \pi)$ for any $\pi \in \Pi_{\text{new}}^{\text{PPO}}$ under all $s_t$. This problem can be formalized as $\pi_{\text{new}}^{\text{PPO}} = argmin_{\pi \in \Pi_{\text{new}}^{\text{PPO}}}(D_{\text{KL}}^{s_1}(\pi_{\text{old}}, \pi), \ldots, D_{\text{KL}}^{s_T}(\pi_{\text{old}}, \pi))$. Note that $\pi(\cdot|s_t)$ is a conditional probability and the optimal solution on different states are independent from each other. Thus the problem can be optimized by independently solving $\min_{\pi(\cdot|s_t) \in \{\pi(\cdot|s_t) : \pi \in \Pi_{\text{new}}^{\text{PPO}}\}} D_{\text{KL}}(\pi_{\text{old}}(\cdot|s_t), \pi(\cdot|s_t))$ for each $s_t$. The final $\pi_{\text{new}}^{\text{PPO}}$ is obtained by integrating these independent optimal solutions $\pi_{\text{new}}^{\text{PPO}}(\cdot|s_t)$ on different state $s_t$. Similarly, $\pi_{\text{new}}^{\text{TRGPPO}}$ is the one of TRGPPO which has similar definition as $\pi_{\text{new}}^{\text{PPO}}$. Please refer to Appendix E for more detail.

To analyse TRGPPO and PPO in a comparable way, we introduce a variant of TRGPPO. The hyperparameter $\delta$ of TRGPPO in eq. (4) is set adaptively by $\epsilon$. That is, $\delta = \max\left((1-p^+)\log\frac{1-p^+}{1-p^+(1+\epsilon)} - p^+\log(1+\epsilon), (1-p^-)\log\frac{1-p^-}{1-p^-(1-\epsilon)} - p^-\log(1-\epsilon)\right)$, where

$p^{+} = \max\limits_{t:A_t>0} \pi_{\text{old}}(a_t|s_t)$, $p^{-} = \max\limits_{t:A_t<0} \pi_{\text{old}}(a_t|s_t)$. One may note that this equation has a similar form to that of eq. (5). In fact, if TRGPPO and PPO share a similar $\epsilon$, then they have the same KL divergence theoretically. We conclude the comparison between TRGPPO and PPO by the following theorem.

**Theorem 4.** *Assume that* $\max_t D_{\text{KL}}^{s_t}(\pi_{\text{old}}, \pi_{\text{new}}^{\text{PPO}}) < +\infty$ *for all t. If TRGPPO and PPO have the same hyperparameter $\epsilon$, we have:*

*(i)* $u_{s_t,a_t}^{\delta} \geq 1 + \epsilon$ *and* $l_{s_t,a_t}^{\delta} \leq 1 - \epsilon$ *for all* $(s_t, a_t)$;

*(ii)* $\max_t D_{\text{KL}}^{s_t}(\pi_{\text{old}}, \pi_{\text{new}}^{\text{TRGPPO}}) = \max_t D_{\text{KL}}^{s_t}(\pi_{\text{old}}, \pi_{\text{new}}^{\text{PPO}})$;

*(iii)* $\hat{M}_{\pi_{\text{old}}}(\pi_{\text{new}}^{\text{TRGPPO}}) \geq \hat{M}_{\pi_{\text{old}}}(\pi_{\text{new}}^{\text{PPO}})$. *Particularly, if there exists at least one* $(s_t, a_t)$ *such that* $\pi_{\text{old}}(a_t|s_t) \neq \max\limits_{\hat{t}:A_{\hat{t}}<0} \pi_{\text{old}}(a_{\hat{t}}|s_{\hat{t}})$ *and* $\pi_{\text{old}}(a_t|s_t) \neq \max\limits_{\hat{t}:A_{\hat{t}}>0} \pi_{\text{old}}(a_{\hat{t}}|s_{\hat{t}})$, *then* $\hat{M}_{\pi_{\text{old}}}(\pi_{\text{new}}^{\text{TRGPPO}}) > \hat{M}_{\pi_{\text{old}}}(\pi_{\text{new}}^{\text{PPO}})$.

Conclusion (i) implies that TRGPPO could enlarge the clipping ranges compared to PPO and accordingly allow larger update of the policy. Meanwhile, the maximum KL divergence is retained, which means TRGPPO would not harm the stability of PPO theoretically. Conclusion (iii) implies that TRGPPO has better empirical performance bound.

## 6 Experiment

We conducted experiments to answer the following questions: (1) Does PPO suffer from the *lack of exploration* issue? (2) Could our TRGPPO relief the exploration issue and improve sample efficiency compared to PPO? (3) Does our TRGPPO maintain the stable learning property of PPO? To answer these questions, we first evaluate the algorithms on two simple bandit problems and then compare them on high-dimensional benchmark tasks.

### 6.1 Didactic Example: Bandit Problems

We first evaluate the algorithms on the bandit problems. In the continuous-armed bandit problem, the reward is $0.5$ for $a \in (1, 2)$; $1$ for $a \in (2.5, 5)$; and $0$ otherwise. We use a Gaussian policy. The discrete-armed bandit problem is defined in Section 4. We use a Gibbs policy $\pi(a) \propto \exp(\theta_a)$, where the parameter $\theta$ is initialized randomly from $\mathcal{N}(0, 1)$. We also consider the vanilla Policy Gradient method as a comparison. Each algorithm was run for 1000 iterations with 10 random seeds.

Fig. 2 plots the performance during the training process. PPO gets trapped in local optima at a rate of 30% and 20% of all the trials on discrete and continuous cases respectively, while our TRGPPO could find the optimal solution on almost all trials. For continuous-armed problem, we have also tried other types of parametrized policies like Beta and Mixture Gaussian, and these policies behaves similarly as the Gaussian policy. In discrete-armed problem, we find that when the policy is initialized with a local optima, PPO could easily get trapped in that one. Notably, since vanilla PG could also find the optimal one, it could be inferred that the exploration issue mainly derives from the ratio-based clipping with constant clipping range.

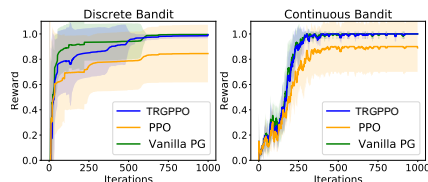

Figure 2: The performance on discrete and continuous-armed bandit problems during training process.

### 6.2 Evaluation on Benchmark Tasks

We evaluate algorithms on benchmark tasks implemented in OpenAI Gym [2], simulated by MuJoCo [21] and Arcade Learning Environment [1]. For continuous control tasks, we evaluate algorithms on 6 benchmark tasks. All tasks were run with 1 million timesteps except that the Humanoid task was 20 million timesteps. The trained policies are evaluated after sampling every 2048 timesteps data. The experiments on discrete control tasks are detailed in Appendix C.

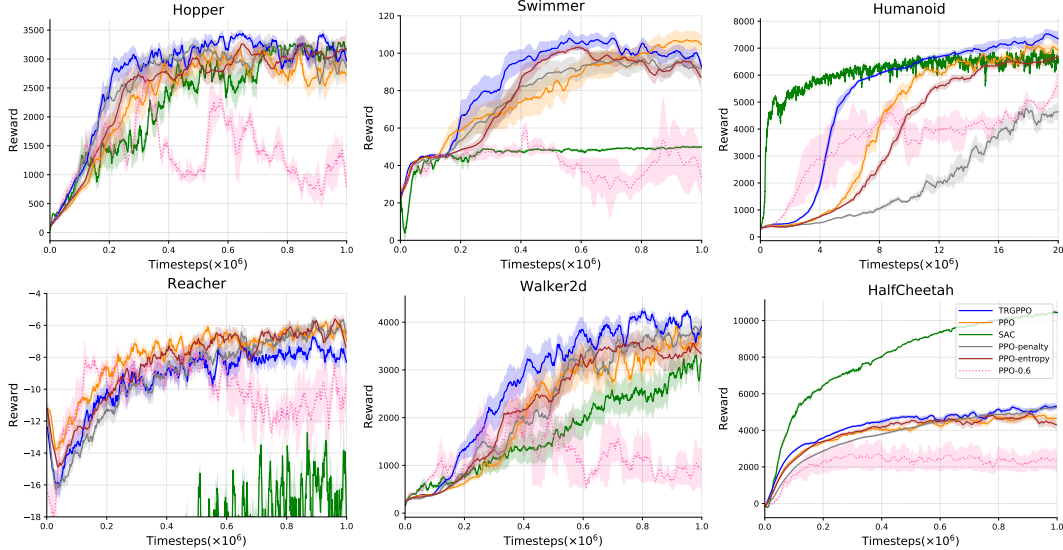

Figure 3: Episode rewards during the training process; the shaded area indicate half the standard deviation over 10 random seeds.

Table 1: Results of timesteps to hit a threshold within 1 million timesteps (except Humanoid with 20 million) and averaged rewards over last 40% episodes during training process.

|  | | (a) Timesteps to hit threshold ($\times 10^3$) | | | | (b) Averaged rewards | | | |
| --- | --- | --- | --- | --- | --- | --- | --- | --- | --- |
|  | Threshold | TRGPPO | PPO | PPO-penalty | SAC | TRGPPO | PPO | PPO-penalty | SAC |
| Humanoid | 5000 | 4653 | 7241 | 13096.0 | **343.0** | **7074.9** | 6620.9 | 3612.3 | 6535.9 |
| Reacher | -5 | 201 | **178.0** | 301.0 | 265 | -7.9 | **-6.7** | -6.8 | -17.2 |
| Swimmer | 90 | **353.0** | 564 | 507.0 | /[4] | **101.9** | 100.1 | 94.1 | 49 |
| HalfCheetah | 3000 | 117 | 148 | 220.0 | **53.0** | 4986.1 | 4600.2 | 4868.3 | **9987.1** |
| Hopper | 3000 | **168.0** | 267 | 188.0 | 209 | **3200.5** | 2848.9 | 3018.7 | 3020.7 |
| Walker2d | 3000 | **269.0** | 454 | 393.0 | 610 | **3886.8** | 3276.2 | 3524 | 2570 |

For our *TRGPPO*, the trust region coefficient $\delta$ is adaptively set by tuning $\epsilon$ (see Appendix B.4 for more detail). We set $\epsilon = 0.2$, same as PPO. The following algorithms were considered in the comparison. (a) *PPO*: we used $\epsilon = 0.2$ as recommended by [18]. (b) *PPO-entropy*: PPO with an explicit entropy regularization term $\beta \mathbb{E}_s \left[ H \left( \pi_{\text{old}}(\cdot|s), \pi(\cdot|s) \right) \right]$, where $\beta = 0.01$. (c) *PPO*-0.6: PPO with a larger clipping range where $\epsilon = 0.6$. (d) *PPO-penalty*: a variant of PPO which imposes a penalty on the KL divergence and adaptively adjust the penalty coefficient [18]. (e) *SAC*: Soft Actor-Critic, a state-of-the-art off-policy RL algorithm [9]. Both TRGPPO and PPO adopt exactly same implementations and hyperparameters except the clipping range based on OpenAI Baselines [4]. This ensures that the differences are due to algorithm changes instead of implementations or hyperparameters. For SAC, we adopt the implementations provided in [9].

**Sample Efficiency:** Table 1 (a) lists the timesteps required by algorithms to hit a prescribed threshold within 1 million timesteps and Figure 3 shows episode rewards during the training process. The thresholds for all tasks were chosen according to [23]. As can be seen in Table 1, TRGPPO requires about only 3/5 timesteps of PPO on 4 tasks except HalfCheetah and Reacher.

**Performance/Exploration:** Table 1 (b) lists the averaged rewards over last 40% episodes during training process. TRGPPO outperforms the original PPO on almost all tasks except Reacher. Fig. 4a shows the policy entropy during training process, the policy entropy of TRGPPO is obviously higher than that of PPO. These results implies that our TRGPPO method could maintain a level of entropy learning and encourage the policy to explore more.

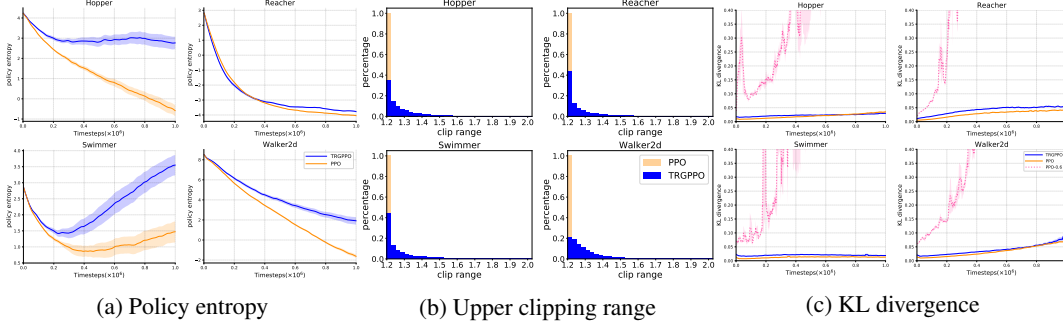

(a) Policy entropy      (b) Upper clipping range      (c) KL divergence

Figure 4: (a) shows the policy entropy during training process. (b) shows the statistics of the computed upper clipping ranges over all samples. (c) shows the KL divergence during the training process.

**The Clipping Ranges and Policy Divergence:** Fig. 4b shows the statistics of the upper clipping ranges of TRGPPO and PPO. Most of the resulted adaptive clipping ranges of TRGPPO are much larger that of PPO. Nevertheless, our method has similar KL divergences with PPO (see Fig. 4c). However, the method of arbitrary enlarging clipping range (PPO-0.6) does not enjoy such property and fails on most of tasks.

**Training Time:** Within one million timesteps, the training wall-clock time for our TRGPPO is 25 min; for PPO, 24 min; for SAC, 182 min (See Appendix B.3 for the detail of evaluation). TRGPPO does not require much additional computation time than PPO does.

**Comparison with State-of-the-art Method:** TRGPPO achieves higher reward than SAC on 5 tasks while is not as good as it on HalfCheetah. And TRGPPO is not as sample efficient as SAC on HalfCheetah and Humanoid. This may due to that TRGPPO is an on-policy algorithm while SAC is an off-policy one. However, TRGPPO is much more computationally efficient (25 min vs. 182 min). In addition, SAC tuned hyperparameters specifically for each task in the implementation of the original authors. In contrast, our TRGPPO uses the same hyperparameter across different tasks.

## 7 Conclusion

In this paper, we improve the original PPO by an adaptive clipping mechanism with a trust region-guided criterion. Our TRGPPO method improves PPO with more exploration and better sample efficiency and is competitive with several state-of-the-art methods, while maintains the stable learning property and simplicity of PPO.

To our knowledge, this is the first work to reveal the effect of the metric of policy constraint on the exploration behavior of the policy learning. While recent works devoted to introducing inductive bias to guide the policy behavior, e.g., maximum entropy learning [24, 8], curiosity-driven method [13]. In this sense, our adaptive clipping mechanism is a novel alternative approach to incorporate prior knowledge to achieve fast and stable policy learning. We hope it will inspire future work on investigating more well-defined policy metrics to guide efficient learning behavior.

## Acknowledgement

This work is partially supported by National Science Foundation of China (61976115,61672280, 61732006), AI+ Project of NUAA(56XZA18009), Postgraduate Research & Practice Innovation Program of Jiangsu Province (KYCX19_0195). We would also like to thank Yao Li, Weida Li, Xin Jin, as well as the anonymous reviewers, for offering thoughtful comments and helpful advice on earlier versions of this work.

## Footnotes

[1]There is also a variant of PPO which uses KL divergence penalty. In this paper we refer to the one clipping probability ratio as PPO by default, which performs better in practice.

[2]$\hat{\pi}_{t+1}$ may violate the probability rules, e.g., $\sum_a \hat{\pi}_{t+1}(a) > 1$. Thus we need to enforce specific normalization operation to rectify it. To simplify the analysis, we assume that $\pi_{t+1} = \hat{\pi}_{t+1}$.

[3]Assume that there is only one optimal action.

[4] '/' means that the method did not reach the reward threshold within the required timesteps on all the seeds.

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
