[Supplementary Material]

# A  Example: Continuous-Armed Bandit

In this section, we show how PPO and TRGPPO perform in continuous action space by a simple continuous-armed bandit problem. Fig. 2 (b) shows the plot of reward function (black dashed curve). Let $a_{\text{subopt}}$ denote any sub-optimal action which achieves second-highest reward; and let $a_{\text{opt}}$ denote the optimal one. The policy is a parametrized Gaussian (orange solid curve).

Fig. 1 shows the clipping ranges for different actions in continuous action space, where $dim(\mathcal{A}) = 1$. The old policy is $\pi_{\text{old}}(a) = \mathcal{N}(a|0, 1)$ (black curve). Note that the probability $\pi_{\text{old}}(a)$ goes smaller as $a$ is away from zero which the mode of the Gaussian distribution. As the figure shows, in continuous action space, our TRGPPO method sets larger clipping range for action which is less likely to be chosen, while PPO sets a constant clipping range under all actions.

Figure 1: Clipping Range of TRGPPO (blue curve) and PPO (orange curve) on different actions. For TRGPPO, $\delta = 0.05$, while for PPO, $\epsilon = 0.2$. The distribution of old policy is $\pi_{\text{old}}(a) = \mathcal{N}(a|0, 1)$.

Fig. 2 shows the training process of PPO on the continuous-armed bandit problem. As can be seen, the allowable improvement of $\pi(a_{\text{opt}})$ of PPO is quite limited. It will require quite a large number of steps for $\pi(a_{\text{opt}})$ to peak. On the other hand, the limited improvement at $\pi(a_{\text{opt}})$ will prevent the policy from allocating more probability at $a_{\text{opt}}$. In other words, it will explore less at $a_{\text{opt}}$. Whereas the allowable improvement at $a_{\text{subopt}}$ is relatively relax. This uneven restriction may result in a growing improvement of $\pi(a_{\text{subopt}})$ and a diminishment of $\pi(a_{\text{opt}})$. Consequently, the policy is trapped in local optima, as Fig. 2 (c) shows. Note that although we use Gaussian distribution as our policy in the example, these issues could also happen on multimodal distribution like Mixture Gaussian or heavy tailed distribution like Beta distribution.

Fig. 3 shows the training process of TRGPPO. When the policy enters into a locally optimal one, the corresponding feasible variation range of $\pi(a_{\text{opt}})$ is close to that of $\pi(a_{\text{subopt}})$ in TRGPPO, as Fig. 3 (b) shows. Note that the advantage value at $a_{\text{opt}}$ is larger than that at $a_{\text{subopt}}$. This could result in a growing reinforcement of $\pi(a_{\text{opt}})$, which accordingly leads to a diminishment of $\pi(a_{\text{subopt}})$. In this way, the policy jumps out of the local optima and converges to the optimal policy, as Fig. 3 (c) shows.

(a) The beginning of training process (1-th iteration)  (b) The middle of training process (200-th iteration)  (c) The end of training process (400-th iteration)

Figure 2: Training process of PPO. The orange solid curve plots the probability density function (PDF) of policy at the training process. The black dashed curve plots the reward function of the bandit problem.

(a) The beginning of training process (1-th iteration)  (b) The middle of training process (200-th iteration)  (c) The end of training process (400-th iteration)

Figure 3: Training process of TRGPPO.

# B Computation of Adaptive Clipping Range

In this section, we detail the method for adaptive clipping range computation, which is formalized as the following problem.

$$l^{\delta}_{s_t,a_t} = \min_{\pi} \left\{ \frac{\pi(a_t|s_t)}{\pi_{\text{old}}(a_t|s_t)} : D^{s_t}_{KL}(\pi_{\text{old}}, \pi) \leq \delta \right\} \tag{1}$$

$$u^{\delta}_{s_t,a_t} = \max_{\pi} \left\{ \frac{\pi(a_t|s_t)}{\pi_{\text{old}}(a_t|s_t)} : D^{s_t}_{KL}(\pi_{\text{old}}, \pi) \leq \delta \right\} \tag{2}$$

To be abbreviated, we describe the approach for minimization case under discrete and continuous action space respectively, while that for the maximization case is similar.

## B.1 Discrete Action Space

For discrete action space tasks, it is standard to use a DNN with softmax output layer to represent the policy, i.e., $\pi(a|s) = [f^p_\theta(s)]_a$, where $f^p_\theta(s)$ is the parameter of categorical distribution on state $s$ and the subscript $a$ denote the $a$-th entry of the vector. The optimal clipping range should be computed should be independent of special parametrization of $f^p_\theta$. Thus the problem is formalized as an optimization problem of two Categorical distributions. Given $s_t$ and $a_t$, let $p' = f^p_{\theta'}(s_t)$, where $\theta'$ is the parameter of old policy $\pi_{\text{old}}$, the computation of $l^{\delta}_{s_t,a_t}$ in (1) is formalized as the following optimization problem:

$$\min_{p} \frac{p_{a_t}}{p'_{a_t}}$$
$$\text{s.t.} \sum_{a \in \mathcal{A}} p'_a \log\frac{p'_a}{p_a} \leq \delta, \quad \sum_{a \in \mathcal{A}} p_a = 1, \tag{3}$$
$$0 \leq p_a \leq 1 \text{ for } a \in \mathcal{A}$$

While the optimization problem for $u^{\delta}_{s_t,a_t}$ is the maximization case of problem (3). Let $\lambda$ and $\nu$ be the Lagrangian multipliers.

$$\begin{cases} \lambda\left(-\frac{p'_a}{p_a}\right) + \nu = 0, \quad \text{for } a \neq a_t & \text{(4a)} \\[2ex] \frac{1}{p'_{a_t}} + \lambda\left(-\frac{p'_{a_t}}{p_{a_t}}\right) + \nu = 0 & \text{(4b)} \\[2ex] \lambda\left(\sum_{a \in \mathcal{A}} p'_a \log\frac{p'_a}{p_a} - \delta\right) = 0 & \text{(4c)} \\[2ex] \sum_{a} p_a = 1 & \text{(4d)} \end{cases}$$

By (a)(b), we have $\lambda \neq 0$, since if $\lambda = 0$ then $\nu = 0$ (by (a)), which contradicts (b). Second, by (c) and $\lambda \neq 0$, we have $\sum_{a \in \mathcal{A}} p'_a \log(p'_a/p_a) = \delta$. Third, taking (a) into (d), we have $p'_a/p_a = \nu/\lambda = (1 - p'_{a_t})/(1 - p_{a_t})$ for $a \neq a_t$. Then, taking this equation into $\sum_{a \in \mathcal{A}} p'_a \log(p'_a/p_a) = \delta$, this problem is transformed into solving the following equation w.r.t. $p_{a_t}$.

$$\left(1 - p'_{a_t}\right) \log\frac{1 - p'_{a_t}}{1 - p_{a_t}} - p'_{a_t} \log\frac{p_{a_t}}{p'_{a_t}} = \delta \tag{5}$$

In fact, there are two groups of solution for (5), where $p_{a_t}/p'_{a_t} < 1$ is the one for the minimization case, while $p_{a_t}/p'_{a_t} > 1$ is the one for the maximization case.

## B.2 Continuous Action Space

For continuous action space tasks, it is standard to represent the stochastic policy by a parameterized conditional Gaussian distribution[4, 2], i.e., $\pi(a|s) = \mathcal{N}(a|f^\mu_\theta(s), f^\Sigma_\theta(s))$, where $f^\mu_\theta$ and $f^\Sigma_\theta$ are two

DNNs which output the mean vector and covariance matrix. Note that the optimal clipping range should be computed independent of special parametrization of $f_\theta^\mu$ and $f_\theta^\Sigma$. Thus the problem (1) is formalized as an optimization problem of two Gaussian distributions. Given $(s_t, a_t)$, let $\mu' = f_{\theta'}^\mu(s_t)$ and $\Sigma' = f_{\theta'}^\Sigma(s_t)$, where $\theta'$ is the parameter of old policy $\pi_{\text{old}}$, the computation of $l_{s_t,a_t}^\delta$ in (1) is formalized as the following optimization problem:

$$
\begin{aligned}
\min_{\mu,\Sigma} \frac{1}{2} &\left( -\log|\Sigma| - (\mu - a_t)^\top \Sigma^{-1} (\mu - a_t) + \log|\Sigma'| + (\mu' - a_t)^\top {\Sigma'}^{-1} (\mu' - a_t) \right) \\
\text{s.t.} \frac{1}{2} &\left( \log \left| \Sigma' \Sigma^{-1} \right| + tr\left\{ {\Sigma'}^{-1}\Sigma \right\} + (\mu - \mu')^\top {\Sigma'}^{-1}(\mu - \mu') - D \right) - \delta \le 0
\end{aligned}
\tag{6}
$$

where $\mu \in \mathbb{R}^D$, $\Sigma \in \mathbb{R}^{D \times D}$ is a positive semi-definite matrix, $D = dim(\mathcal{A})$ is the dimension of action space. The objective function is log of the ratio $\pi$. The covariance matrix could be decomposed by $\Sigma' = \bar{\Sigma}'\bar{\Sigma}'^\top$, and we introduce a rotation matrix $R \in \mathbb{R}^{D \times D}$ (which has $R^\top R = R^{-1}R = I$).

Second, by replacing $\mu$ with $\bar{\Sigma}'R\mu + \mu'$, and $\Sigma$ with $\bar{\Sigma}'R\Sigma R^\top \bar{\Sigma}'^\top$, we could transform the problem to

$$
\begin{aligned}
\min_{\mu,\Sigma} \frac{1}{2} &\left( -\log|\Sigma| - (\mu - \bar{a}_t)^\top \Sigma^{-1} (\mu - \bar{a}_t) + \bar{a}_t^\top \bar{a}_t \right) \\
s.t. \frac{1}{2} &\left( \log \left| \Sigma^{-1} \right| + tr\left\{ \Sigma \right\} + \mu^\top \mu - D \right) - \delta \le 0
\end{aligned}
\tag{7}
$$

where $\bar{a}_t = R^\top \bar{\Sigma}'^{-1} (\mu' - a_t)$.

Next, we constrain the covariance matrix $\Sigma$ to be diagonal. The final result is sub-optima compared to the original problem. However, we don't require accurate clipping bound when optimizing policy. Another reason is that in practice the diagonal Gaussian policy is widely used in RL realizations. Then (7) is equivalent to the following problem.

$$
\begin{aligned}
\min_{\mu,\sigma} \frac{1}{2} \sum_{d=1}^{D} &\left( -\log \sigma_d - (\mu_d - \bar{a}_{t,d})^2 \sigma_d^{-1} + \bar{a}_{t,d}^2 \right) \\
\text{s.t.} \frac{1}{2} \sum_{d=1}^{D} &\left[ -\log \sigma_d + \sigma_d + \mu_d^2 - D \right] - \delta \le 0
\end{aligned}
\tag{8}
$$

where $\mu \in \mathbb{R}^D$, $\sigma \in \mathbb{R}^{+D}$. We choose appropriate $R$ to make $\bar{a}_{t,d} = \dot{a}_t = \left\| (\mu' - a_t) \bar{\Sigma}'^{-1} \right\| / \sqrt{D}$ for $d = 1, 2, \cdots, D$, which means that all entries of $\bar{a}_t$ are equal. Let $\lambda$ be the Lagrangian multiplier, by appling the KKD condition,

$$
\begin{cases}
-\frac{(\mu_d - \dot{a}_t)}{\sigma_d} + \lambda \mu_d = 0 & d = 1, \cdots, D \\
-\frac{1}{\sigma_d} + \frac{(\mu_d - \dot{a}_t)^2}{\sigma_d^2} + \lambda \left[ -\frac{1}{\sigma_d} + 1 \right] = 0 & d = 1, \cdots, D \\
\lambda \left( \frac{1}{2} \sum_{d=1}^{D} \left[ -\log \sigma_d + \sigma_d + \mu_d^2 - D \right] - \delta \right) = 0 \\
\lambda \ge 0
\end{cases}
\tag{9}
$$

By the equations above, we could easily know that $\mu_d$ and $\sigma_d$ are equal for all $d$. Thus the problem could collapse to the following problem,

$$
\begin{aligned}
\min_{\mu,\sigma} \frac{1}{2}D &\left( -\log \sigma - (\mu - \dot{a}_t)^2 \sigma^{-1} + (0 - \dot{a}_t)^2 \right) \\
s.t. \frac{1}{2} &\left( -\log \sigma + \sigma + \mu^2 - 1 \right) - D^{-1}\delta \le 0
\end{aligned}
\tag{10}
$$

Remember that $\dot{a}_t = \left\| (\mu' - a_t) \bar{\Sigma}'^{-1} \right\| / \sqrt{D}$, $\mu \in \mathbb{R}$, $\sigma \in \mathbb{R}^+$. Until now, the original $D$-dimensional optimization problem is transformed to a 1-dimensional optimization problem. By the KKT conditions above, we could obtain the following equations w.r.t. $\mu$, $\sigma$ and $\lambda$.

When $\dot{a}_t \neq 0$, the problem is transformed into solving the following equations w.r.t. $\mu, \sigma, \lambda$:

$$\begin{cases} -\log \frac{(\mu - \dot{a}_t)(\mu^2 - \dot{a}_t \mu - 1)}{\dot{a}_t} + \frac{(\mu - \dot{a}_t)(\mu^2 - \dot{a}_t \mu - 1)}{\dot{a}_t} + \mu^2 - 1 - 2\delta D^{-1} = 0 \\ \sigma = (\mu - \dot{a}_t)(\mu^2 - \dot{a}_t \mu - 1)/\dot{a}_t \\ \lambda = \frac{\mu - \dot{a}_t}{\sigma \mu} \end{cases} \quad (11)$$

When $\dot{a}_t = 0$, the problem is transformed into solving the following equations w.r.t. $\mu, \sigma, \lambda$:

$$\begin{cases} \mu = 0 \\ -\log \sigma + \sigma - 1 - 2\delta/D = 0 \\ \lambda = \frac{-\sigma + \dot{a}_t^2}{-\sigma + \sigma^2} \end{cases} \quad (12)$$

There are two groups of solution for both (11) and (12), where $\lambda > 0$ is the one for the minimization case, while $\lambda < 0$ is the one for the maximization case.

## B.3 Computation Acceleration

Note the solutions in (5) only depend on the one-dimensional constant $p'_a$ and $\delta$, while (11) and (12) only depend on one-dimensional constant $\dot{a}$ and $\delta/D$. We use MINPACK's HYBRD and HYBRJ routines [3] as the solver. To accelerate this computation procedure, we propose two additional approach. One is to train a DNN which input $\pi_{\text{old}}(a|s)$ and $\delta$ and approximately output the solutions, which serve as the initial solutions for the solver. The other is to discretize the space of the input and save all the solutions in advance. The experimental results in our main content are conducted with the discretization version.

Table 2 shows the wall-clock time required by variants of TRGPPO and PPO to finish benchmark tasks in a modern CPU. With our proposed acceleration tricks, the optimization time of calculating clipping range can be reduced significantly. The result is obtained with the same experiment setup as previous experiments. The experiments are applied on a computer with an Intel i5-7500 CPU, 16GB of memory and a GeForce GTX 1060 GPU.

Table 1: Input and output of the DNN for solving problems. For discrete action space, we sample 1000 $p'_a$. For continuous action space, we sample 1000 $\dot{a}$ and 1000 $\delta/D$ (note we take $\delta/D$ as an entity). We solve these problems and obtain the corresponding solutions, and these data are used to train our DNN.

| | Input | Output |
|---|---|---|
| **Discrete Action Space** | $p'_a \sim U(0,1)$ | $p_a$ |
| **Continuous Action Space** | $\dot{a} \sim U(-5,5); D^{-1}\delta \sim U[0.0002, 0.01]$ | $\mu, \sigma$ |

Table 2: Comparison of computation cost for TRGPPO with different acceleration tricks.

| | PPO | TRGPPO (discretization) | TRGPPO (solver) | TRGPPO (DNN) |
|---|---|---|---|---|
| **Mujoco($10^6$ timesteps)** | 24 min | 25 min | 52 min | 29 min |
| **Atari($10^7$ timesteps)** | 195 min | 198 min | 243 min | 213 min |

## B.4 Adaptively Setting $\delta$ with $\epsilon$

We detail method about how to adaptively set $\delta$ by $\epsilon$. Our goal is to make TRGPPO has theoretical maximum KL divergence over all sampled states.

For discrete action space, let $p^+ = \max\limits_{t:A_t>0} \pi_{\text{old}}(a_t|s_t)$, $p^- = \max\limits_{t:A_t<0} \pi_{\text{old}}(a_t|s_t)$. By eq. (5), we set $\delta$ by

$$\delta = max(\delta^+, \delta^-) \quad (13a)$$

$$\delta^+ = \left(1 - p^+\right) \log \frac{1 - p^+}{1 - p^+(1 + \epsilon)} - p^+ \log(1 + \epsilon) \quad (13b)$$

$$\delta^- = \left(1 - p^-\right) \log \frac{1 - p^-}{1 - p^-(1 - \epsilon)} - p^- \log(1 - \epsilon) \quad (13c)$$

Figure 4: Episode rewards achieved by algorithm during the training process averaged over 4 random seeds. TRGPPO (blue line) achieves better performance than PPO (orange line).

For continuous action space, for PPO, theoretically, it always achieves maximum KL divergence at the sampled action which is the mode of the Gaussian. Our idea is to make the optimal clipping range at the mode of distribution equals the clipping range of PPO. By problem eq. (10) and (12), we set $\delta$ by

$$\delta = max(\delta^+, \delta^-) \tag{14a}$$

$$\delta^+ = \log(1 + \epsilon) + \frac{1}{2}D\exp(\frac{-2\log(1 + \epsilon)}{D}) - \frac{1}{2}D \tag{14b}$$

$$\delta^- = \log(1 - \epsilon) + \frac{1}{2}D\exp(\frac{-2\log(1 - \epsilon)}{D}) - \frac{1}{2}D \tag{14c}$$

## C  Additional Experiment

To evaluate the proposed TRGPPO on discrete tasks, we use Atari games as a testing environment, so the policies are learned with raw images. We present results on several atari games in Fig. 4, the blue and orange curves visualize the results using TRGPPO and PPO. We set $\delta = 0.001$ for all tasks.

Both TRGPPO and PPO adopt exactly same implementations and hyperparameters given in [1] for discrete tasks except that clipping range of TRGPPO is computed adaptively according to given $\delta$, and the policy entropy coefficient is 0 but not 0.01 used in PPO. This is because our TRGPPO has better *exploration* property than PPO, so it does not need to add extra entropy regularization.

## D  Implementation Details

Table 3: Hyperparameters for PPO and TRGPPO on Mujoco tasks.

| Hyperparameter | Value |
|---|---|
| learning rate | $3 \times 10^{-4}$ |
| number of parallel environments | 64 (Humanoid) 2 (Other tasks) |
| timesteps per epoch | 1024 |
| initial logstd of policy | -1.34 (HalfCheetah,Humanoid) 0 (Other tasks) |
| policy | Gaussian |
| $\lambda$ (GAE) | 0.95 |
| clipping range $\epsilon$ (PPO and TRGPPO) | 0.2 |

Table 4: Hyperparameters for PPO and TRGPPO on Atari tasks.

| Hyperparameter | Value |
|---|---|
| learning rate | $2.5 \times 10^{-4}$ |
| number of parallel environments | 8 |
| timesteps per epoch | 128 |
| policy | Softmax |
| $\lambda$ (GAE) | 0.95 |
| clipping range $\epsilon$ (PPO) | LinearAnneal(0.1,0) |
| coefficient of trust region $\delta$ (TRGPPO) | LinearAnneal(0.001,0) |

# E  Theorem Proof

In this section, we will give theorem proofs. To make it easier to read, we will mention the related notations again.

## E.1  Theorems in Section 4

**Lemma 1.** $\Delta_{\pi_0,t} \triangleq \mathbb{E}_{\pi_t}\left[\|\pi_t - \pi^*\|_\infty | \pi_0\right] = 1 - \mathbb{E}_{\pi_t}\left[\pi_t(a_{\mathrm{opt}}) | \pi_0\right]$.

**Proof:** For any $a \neq a_{\mathrm{opt}}$, then $|\pi_t(a) - \pi^*(a)| = |\pi_t(a)| \leq |1 - \pi_t(a_{\mathrm{opt}})|$.

Thus we have $\|\pi_t - \pi^*\|_\infty = \max_{a \in \mathcal{A}} \|\pi_t(a) - \pi^*(a)\| = 1 - \pi_t(a_{\mathrm{opt}})$.

**Lemma 2.** $\mathbb{E}_{\pi_{t+1}}\left[\pi_{t+1}(a) | \pi_0\right] = \mathbb{E}_{\pi_t}\left[\mathbb{E}_{\pi_{t+1}}\left[\pi_{t+1}(a) | \pi_t\right] | \pi_0\right]$.

**Theorem 2.** *Given initial policy* $\pi_0$, *if* $\pi_0^2(a_{\mathrm{opt}}) \cdot |\mathcal{A}| < \sum_{a_{\mathrm{subopt}} \in \mathcal{A}_{\mathrm{subopt}}} \pi_0^2(a_{\mathrm{subopt}}) - \sum_{a^- \in \mathcal{A}^-} \pi_0^2(a^-)$, *then we have*

*(i)* $\sum_{a_{\mathrm{subopt}} \in \mathcal{A}_{\mathrm{subopt}}} \pi_0(a_{\mathrm{subopt}}) < \sum_{a_{\mathrm{subopt}} \in \mathcal{A}_{\mathrm{subopt}}} \mathbb{E}_{\pi_1^{\mathrm{PPO}}}\left[\pi_1^{\mathrm{PPO}}(a_{\mathrm{subopt}}) | \pi_0\right] < \cdots < \sum_{a_{\mathrm{subopt}} \in \mathcal{A}_{\mathrm{subopt}}} \mathbb{E}_{\pi_t^{\mathrm{PPO}}}\left[\pi_t^{\mathrm{PPO}}(a_{\mathrm{subopt}}) | \pi_0\right]$;

*(ii)* $\pi_0(a_{\mathrm{opt}}) > \mathbb{E}_{\pi_1^{\mathrm{PPO}}}\left[\pi_1^{\mathrm{PPO}}(a_{\mathrm{opt}}) | \pi_0\right] > \cdots > \mathbb{E}_{\pi_t^{\mathrm{PPO}}}\left[\pi_t^{\mathrm{PPO}}(a_{\mathrm{opt}}) | \pi_0\right]$;

*(iii)* $\Delta_{\pi_0,0} < \Delta_{\pi_0,1}^{\mathrm{PPO}} < \cdots < \Delta_{\pi_0,t}^{\mathrm{PPO}}$.

**Proof:** For PPO, if $c(a) > 0$, then we have

$$\mathbb{E}_{\pi_{t+1}^{\mathrm{PPO}}}\left[\pi_{t+1}^{\mathrm{PPO}}(a) | \pi_t\right] = \pi_t(a) + \left[\pi_t^2(a) - \sum_{a^+ \in \mathcal{A}^+/\{a\}} \frac{\pi_t^2(a^+)}{|\mathcal{A}|-1} + \sum_{a^- \in \mathcal{A}^-} \frac{\pi_t^2(a^-)}{|\mathcal{A}|-1}\right]\epsilon \quad (15)$$

Let $L(a) = \pi_t^2(a) - \sum_{a^+ \in \mathcal{A}^+/\{a\}} \frac{\pi_t^2(a^+)}{|\mathcal{A}|-1} + \sum_{a^- \in \mathcal{A}^-} \frac{\pi_t^2(a^-)}{|\mathcal{A}|-1}$.

If $\pi_0^2(a_{\mathrm{opt}}) \cdot |\mathcal{A}| < \sum_{a_{\mathrm{subopt}} \in \mathcal{A}_{\mathrm{subopt}}} \pi_0^2(a_{\mathrm{subopt}}) - \sum_{a^- \in \mathcal{A}^-} \pi_0^2(a^-)$, then we have $L(a_{\mathrm{opt}}) < 0$.

Hence we obtain $\pi_0^{\mathrm{PPO}}(a_{\mathrm{opt}}) > \mathbb{E}_{\pi_1^{\mathrm{PPO}}}\left[\pi_1^{\mathrm{PPO}}(a_{\mathrm{opt}}) | \pi_t\right]$.

For $a_{\mathrm{subopt}}$, we have

$$\sum_{a \in \mathcal{A}_{\mathrm{subopt}}} L(a_{\mathrm{subopt}})$$
$$= \sum_{a \in a_{\mathrm{subopt}}} \pi_0^2(\mathrm{subopt}) - \pi_0^2(a^+)\frac{|\mathcal{A}_{\mathrm{subopt}}|}{|\mathcal{A}|-1} + \sum_{a^- \in \mathcal{A}^-} \frac{\pi_0^2(a^-)}{|\mathcal{A}|-1} > 0 \quad (16)$$

Thus we have

$$\sum_{a_{\mathrm{subopt}} \in \mathcal{A}_{\mathrm{subopt}}} \pi_0(a_{\mathrm{subopt}}) < \sum_{a_{\mathrm{subopt}} \in \mathcal{A}_{\mathrm{subopt}}} \mathbb{E}_{\pi_1^{\mathrm{PPO}}}\left[\pi_1^{\mathrm{PPO}}(a_{\mathrm{subopt}}) | \pi_0\right] \quad (17)$$

Then by Lemma 2, we obtain (i) and (ii). Since (ii) holds, by Lemma 1, we get (iii).

□

## E.2 Theorems in Section 5

**Lemma 3.** *For TRGPPO with hyperparameter $\delta$, we have $\frac{du^\delta_{s,a}}{d\pi_{\text{old}}(a|s)} < 0$, $\frac{dl^\delta_{s,a}}{d\pi_{\text{old}}(a|s)} > 0$.*

**Proof:** By solving (1) and (2) for discrete space, we have

$$(1 - \pi_{\text{old}}(a|s)) \log \frac{1 - \pi_{\text{old}}(a|s)}{1 - \pi_{\text{old}}(a|s)l^\delta_{s,a}} - \pi_{\text{old}}(a|s) \log l^\delta_{s,a} = \delta \tag{18}$$

$$(1 - \pi_{\text{old}}(a|s)) \log \frac{1 - \pi_{\text{old}}(a|s)}{1 - \pi_{\text{old}}(a|s)u^\delta_{s,a}} - \pi_{\text{old}}(a|s) \log u^\delta_{s,a} = \delta \tag{19}$$

To be abbreviated, let $l = l^\delta_{s_t,a_t}$, $u = u^\delta_{s_t,a_t}$, $p = \pi_{\text{old}}(a_t|s_t)$. By eq. (18), we have

$$\begin{aligned}
\frac{dl}{dp} &= \frac{l\left((1-pl)\log\frac{(1-p)l}{1-pl} + 1 - l\right)}{p(l-1)} \\
&= \frac{l(1-pl)}{p(l-1)}\left(\log\frac{(1-p)l}{1-pl} + \frac{1-l}{1-pl}\right) \\
&= \frac{l(1-pl)}{p(l-1)}\left(\log\left(1 + \frac{l-1}{1-pl}\right) - \frac{l-1}{1-pl}\right)
\end{aligned} \tag{20}$$

Note that $0 < l < 1$, $1 - pl > 0$, we have $\frac{l-1}{1-pl} = -\frac{1-l}{1-pl} > -1$. Indeed, $\log(1+x) - x < 0$ for any $x > -1$. Hence, $\log\left(1 + \frac{l-1}{1-pl}\right) - \frac{l-1}{1-pl} < 0$. We obtain $\frac{dl}{dp} > 0$.

Similarly, we can get that $\frac{du}{dp} > 0$.

□

**Theorem 3.** *For TRGPPO with hyperparameter $(\delta, \epsilon)$ and PPO with same $\epsilon$. If $\delta \leq g(\max_{a \in \mathcal{A}_{\text{subopt}}} \pi_t(a), 1 + \epsilon)$ for all $t$, then we have $\Delta^{\text{TRGPPO}}_{\pi_0,t} \leq \Delta^{\text{PPO}}_{\pi_0,t}$ for any $t$.*

**Proof:** If $\delta \leq g(\max_{a \in \mathcal{A}_{\text{subopt}}} \pi_t(a), 1 + \epsilon)$, then by Lemma 3 and Lemma 5 we have $u^\delta_{a_{\text{subopt}}} \leq 1 + \epsilon$. Hence, $u^{\delta,\epsilon}_{a_{\text{subopt}}} = 1 + \epsilon$.

Meanwhile, $u^{\delta,\epsilon}_a \geq 1 + \epsilon$ and $l^{\delta,\epsilon}_a \leq 1 - \epsilon$ for any $a$.

If $c(a) \geq 0$, then we have

$$\mathbb{E}_{\pi_{\text{new}}}[\pi_{\text{new}}(a)|\pi_t] = \pi_t(a) + \left[\pi_t^2(a)(u_a - 1) - \sum_{a^+ \in \mathcal{A}^+/\{a\}} \frac{\pi_t^2(a^+)}{|\mathcal{A}|-1}(u_{a^+} - 1) + \sum_{a^- \in \mathcal{A}^-} \frac{\pi_t^2(a^-)}{|\mathcal{A}|-1}(1 - l_{a^-})\right] \tag{21}$$

Since $u^{\delta,\epsilon}_{a_{\text{opt}}} \geq 1 + \epsilon$ and $l^{\delta,\epsilon}_{a^-} \leq 1 - \epsilon$ while $u^{\delta,\epsilon}_{a_{\text{subopt}}} = 1 + \epsilon$, we can get $\mathbb{E}_{\pi^{\text{TRGPPO}}_{t+1}}\left[\pi^{\text{TRGPPO}}_{t+1}(a_{\text{opt}})|\pi_t\right] \geq \mathbb{E}_{\pi^{\text{PPO}}_{t+1}}\left[\pi^{\text{PPO}}_{t+1}(a_{\text{opt}})|\pi_t\right]$.

Then by Lemma 2, we have $\mathbb{E}_{\pi^{\text{TRGPPO}}_{t+1}}\left[\pi^{\text{TRGPPO}}_{t+1}(a_{\text{opt}})|\pi_0\right] \geq \mathbb{E}_{\pi^{\text{PPO}}_{t+1}}\left[\pi^{\text{PPO}}_{t+1}(a_{\text{opt}})|\pi_0\right]$.

Finally, by Lemma 1, we have $\Delta^{\text{TRGPPO}}_{\pi_0,t} \leq \Delta^{\text{PPO}}_{\pi_0,t}$.

□

We now derive the form of the optimal solution which achieves minimum KL divergence over all optimal solutions. The general form of surrogate objective function of PPO is as follows:

$$\hat{L}_{\pi_{\text{old}}}^{\text{CLIP}}(\pi) = \frac{1}{T} \sum_{t=1}^{T} \left[ \min \left( r_\pi(s_t, a_t) A_t, clip \left( r_\pi(s_t, a_t), l_{s_t,a_t}, u_{s_t,a_t} \right) A_t \right) \right] \tag{22}$$

Let $\Pi_{\text{new}}$ denote the set of all the optimal solutions of the empirical surrogate objective function of PPO, and let $\pi_{\text{new}} \in \Pi_{\text{new}}$ denote the optimal solution which achieves minimum KL divergence over all optimal solutions, i.e., $D_{\text{KL}}^{s_t}(\pi_{\text{old}}, \pi_{\text{new}}) \leq D_{\text{KL}}^{s_t}(\pi_{\text{old}}, \pi)$ for any $\pi \in \Pi_{\text{new}}$ under all $s_t$.

### We first give the form of $\Pi_{\text{new}}$.

**Lemma 4.** $\Pi_{\text{new}} = \{\pi | for\ all\ t\ that\ A_t < 0, \pi(a_t|s_t) \leq \pi_{\text{old}}(a_t|s_t) l_{s_t,a_t}; for\ all\ t\ that\ A_t > 0, \pi(a_t|s_t) \geq \min(\pi_{\text{old}}(a_t|s_t) u_{s_t,a_t}, 1)\}.$

**Proof:**

We first prove that if a policy $\pi^*$ satisfies the conditions in $\Pi_{\text{new}}$, then $\pi^* \in \Pi_{\text{new}}$.

Let $\hat{L}_{\pi_{\text{old}}}^t(\pi) = \min \left( r_\pi(s_t, a_t) A_t, clip \left( r_\pi(s_t, a_t), l_{s_t,a_t}, u_{s_t,a_t} \right) A_t \right)$. To prove that $\hat{L}_{\pi_{\text{old}}}^{\text{CLIP}}(\pi^*) \geq \hat{L}_{\pi_{\text{old}}}^{\text{CLIP}}(\pi)$ for any $\pi$, we just need to prove that $\hat{L}_{\pi_{\text{old}}}^t(\pi^*) \geq \hat{L}_{\pi_{\text{old}}}^t(\pi)$ for any $\pi$ under all $t$.

If $A_t < 0$, $\hat{L}_{\pi_{\text{old}}}^t(\pi)$ could be rewritten as the following form:

$$\hat{L}_{\pi_{\text{old}}}^t(\pi) = \begin{cases} l_{s_t,a_t} A_t & r_\pi(s_t, a_t) \leq l_{s_t,a_t} \\ r_\pi(s_t, a_t) A_t & r_\pi(s_t, a_t) > l_{s_t,a_t} \end{cases} \tag{23}$$

Thus, we have $\hat{L}_{\pi_{\text{old}}}^t(\pi) \leq l_{s_t,a_t} A_t = \hat{L}_{\pi_{\text{old}}}^t(\pi^*)$ for any $\pi$.

Similarly, if $A_t > 0$, we also have $\hat{L}_{\pi_{\text{old}}}^t(\pi) \leq \hat{L}_{\pi_{\text{old}}}^t(\pi^*)$ for any $\pi$.

We then prove that if a policy $\pi_0$ does not satisfy the conditions in $\Pi_{\text{new}}$, then $\pi^*$ is not an optimal solution in maximization problem of eq. (22).

We can construct a policy $\pi^*$ that satisfy the conditions in the $\Pi_{\text{new}}$. We have $\hat{L}_{\pi_{\text{old}}}^t(\pi_0) < \hat{L}_{\pi_{\text{old}}}^t(\pi^*)$ on $t$ that does not satisfy the conditions. Hence, $\hat{L}_{\pi_{\text{old}}}^{\text{CLIP}}(\pi_0) < \hat{L}_{\pi_{\text{old}}}^{\text{CLIP}}(\pi^*)$.

$\square$

### We now derive the form of $\pi_{\text{new}}$.

If $A_t < 0$, by Lemma 4, $\min_{\pi \in \Pi_{\text{new}}} D_{\text{KL}}^{s_t}(\pi_{\text{old}}, \pi)$ is formalized as the following problem:

$$\min_{\pi} \sum_a \pi_{\text{old}}(a|s_t) \log \frac{\pi_{\text{old}}(a_t|s_t)}{\pi(a_t|s_t)}$$
$$s.t. \pi(a_t|s_t) \leq \pi_{\text{old}}(a_t|s_t) l_{s_t,a_t}, \tag{24}$$
$$\sum_a \pi(a|s_t) = 1, \pi(a|s_t) > 0$$

By using the KKT conditions, we can get that

$$\pi_{\text{new}}(a|s_t) = \begin{cases} \frac{\pi_{\text{old}}(a|s_t)(1 - \pi_{\text{old}}(a_t|s_t) l_{s_t,a_t})}{1 - \pi_{\text{old}}(a_t|s_t)} & a \neq a_t \\ \pi_{\text{old}}(a_t|s_t) l_{s_t,a_t} & a = a_t \end{cases} \tag{25}$$

The according KL divergence is

$$D_{\text{KL}}^{s_t}(\pi_{\text{old}}, \pi_{\text{new}}) = (1 - \pi_{\text{old}}(a_t|s_t)) \log \frac{1 - \pi_{\text{old}}(a_t|s_t)}{1 - \pi_{\text{old}}(a_t|s_t) l_{s_t,a_t}} - \pi_{\text{old}}(a_t|s_t) \log l_{s_t,a_t} \tag{26}$$

Similarly, if $A_t > 0$, we can get

$$\pi_{\text{new}}(a|s_t) = \begin{cases} \frac{\pi_{\text{old}}(a|s_t)(1 - \min(\pi_{\text{old}}(a_t|s_t) u_{s_t,a_t}, 1))}{1 - \pi_{\text{old}}(a_t|s_t)} & a \neq a_t \\ \min(\pi_{\text{old}}(a_t|s_t) u_{s_t,a_t}, 1) & a = a_t \end{cases} \tag{27}$$

If $A_t > 0$ and $\pi_{\text{old}}(a_t|s_t)u_{s_t,a_t} \leq 1$, the according KL is

$$D_{\text{KL}}^{s_t}(\pi_{\text{old}}, \pi_{\text{new}}) = (1 - \pi_{\text{old}}(a_t|s_t))\log\frac{1 - \pi_{\text{old}}(a_t|s_t)}{1 - \pi_{\text{old}}(a_t|s_t)u_{s_t,a_t}} - \pi_{\text{old}}(a_t|s_t)\log u_{s_t,a_t} \quad (28)$$

If $A_t > 0$ and $\pi_{\text{old}}(a_t|s_t)u_{s_t,a_t} > 1$, we have $D_{\text{KL}}^{s_t}(\pi_{\text{old}}, \pi_{\text{new}}) = +\infty$. Equation (26) and eq. (28) have just the same form w.r.t. $l_{s_t,a_t}$ and $u_{s_t,a_t}$ respectively. In fact, since $l_{s_t,a_t} \in (0,1)$ and $u_{s_t,a_t} \in (1, +\infty)$, the monotonicity w.r.t. $l_{s_t,a_t}$ and $u_{s_t,a_t}$ on these two intervals are different, and we obtain the correlation between clipping range and KL divergence.

**Lemma 5.** *(i) If $A_t < 0$, we have $dD_{\text{KL}}^{s_t}(\pi_{\text{old}}, \pi_{\text{new}})/dl_{s_t,a_t} < 0$, $dD_{\text{KL}}^{s_t}(\pi_{\text{old}}, \pi_{\text{new}})/d\pi_{\text{old}}(a_t|s_t) > 0$. (ii) If $A_t > 0$ and $\pi_{\text{old}}(a_t|s_t)u_{s_t,a_t} \leq 1$, we have $dD_{\text{KL}}^{s_t}(\pi_{\text{old}}, \pi_{\text{new}})/du_{s_t,a_t} > 0$, $dD_{\text{KL}}^{s_t}(\pi_{\text{old}}, \pi_{\text{new}})/d\pi_{\text{old}}(a_t|s_t) > 0$.*

**Proof:** To be abbreviated, let $D = D_{\text{KL}}^{s_t}(\pi_{\text{old}}, \pi_{\text{new}})$, $l = l_{s_t,a_t}^\delta$, $u = u_{s_t,a_t}^\delta$, $p = \pi_{\text{old}}(a_t|s_t)$.

If $A_t < 0$, by eq. (26), we have

$$\begin{aligned}
\frac{dD}{dp} &= -\log\frac{(1-p)l}{1-pl} + \frac{l-1}{1-pl} \\
&= -\log\left(1 + \frac{l-1}{1-pl}\right) + \frac{l-1}{1-pl}
\end{aligned} \quad (29)$$

If $A_t > 0$ and $\pi_{\text{old}}(a_t|s_t)u_{s_t,a_t} \leq 1$, by eq. (28), we have

$$\frac{dD}{dp} = -\log\left(1 + \frac{u-1}{1-pu}\right) + \frac{u-1}{1-pu} \quad (30)$$

We have $\frac{l-1}{1-pl} > -1$ and $\frac{u-1}{1-pu} > 0$. Indeed, $-\log(1+x) + x > 0$ for any $x > -1$. Thus, we have $\frac{dD}{dp} > 0$.

If $A_t < 0$, by eq. (26), we have

$$\frac{dD}{dl} = \frac{p(l-1)}{l(1-pl)} < 0 \quad (31)$$

If $A_t > 0$ and $\pi_{\text{old}}(a_t|s_t)u_{s_t,a_t} \leq 1$, by eq. (28), we have

$$\frac{dD}{du} = \frac{p(u-1)}{u(1-pu)} > 0 \quad (32)$$

$\square$

We introduce an empirical version of lower performance bound.

$$\hat{M}_{\pi_{\text{old}}}(\pi) = \hat{L}_{\pi_{\text{old}}}(\pi) - C\max_t D_{\text{KL}}^{s_t}(\pi_{\text{old}}, \pi). \quad (33)$$

where $\hat{L}_{\pi_{\text{old}}}(\pi) = \frac{1}{T}\sum_{t=1}^{T}[r_\pi(s_t, a_t)A_t] + \hat{\eta}^{\pi_{\text{old}}}$, $\hat{\eta}^{\pi_{\text{old}}}$ is the estimated performance of $\pi_{\text{old}}$.

**Lemma 6.** *(i) For PPO, assume that $\max_t D_{\text{KL}}^{s_t}(\pi_{\text{old}}, \pi_{\text{new}}^{\text{PPO}}) < +\infty$, if a given $(s_t, a_t)$ satisfies $\pi_{\text{old}}(a_t|s_t) \neq \max_{\hat{t}:A_{\hat{t}}<0}\pi_{\text{old}}(a_{\hat{t}}|s_{\hat{t}})$ and $\pi_{\text{old}}(a_t|s_t) \neq \max_{\hat{t}:A_{\hat{t}}>0}\pi_{\text{old}}(a_{\hat{t}}|s_{\hat{t}})$, then $D_{\text{KL}}^{s_t}(\pi_{\text{old}}, \pi_{\text{new}}^{\text{PPO}}) < \max_{\hat{t}} D_{\text{KL}}^{s_{\hat{t}}}(\pi_{\text{old}}, \pi_{\text{new}}^{\text{PPO}})$. (ii) For TRGPPO, we have $D_{\text{KL}}^{s_t}(\pi_{\text{old}}, \pi_{\text{new}}^{\text{TRGPPO}}) = \max_{\hat{t}} D_{\text{KL}}^{s_{\hat{t}}}(\pi_{\text{old}}, \pi_{\text{new}}^{\text{TRGPPO}})$ for any $(s_t, a_t)$.*

**Proof:** We first prove (i).

If $A_t < 0$, if $\pi_{\text{old}}(a_t|s_t) \neq \max_{\hat{t}:A_{\hat{t}}<0}\pi_{\text{old}}(a_{\hat{t}}|s_{\hat{t}})$, then $\pi_{\text{old}}(a_t|s_t) < \max_{\hat{t}:A_{\hat{t}}<0}\pi_{\text{old}}(a_{\hat{t}}|s_{\hat{t}})$. By Theorem 5, we have $D_{\text{KL}}^{s_t}(\pi_{\text{old}}, \pi_{\text{new}}^{\text{PPO}}) < \max_{\hat{t}:A_{\hat{t}}<0} D_{\text{KL}}^{s_{\hat{t}}}(\pi_{\text{old}}, \pi_{\text{new}}^{\text{PPO}}) \leq \max_{\hat{t}} D_{\text{KL}}^{s_{\hat{t}}}(\pi_{\text{old}}, \pi_{\text{new}}^{\text{PPO}})$.

Similarly, if $A_t > 0$, we also have $D_{\text{KL}}^{s_t}(\pi_{\text{old}}, \pi_{\text{new}}^{\text{PPO}}) < \max_{\hat{t}} D_{\text{KL}}^{s_{\hat{t}}}(\pi_{\text{old}}, \pi_{\text{new}}^{\text{PPO}})$.

We then prove (ii). If $A_t < 0$, by eq. (18) and eq. (26), we have

$$D_{\mathrm{KL}}^{s_t}(\pi_{\mathrm{old}}, \pi_{\mathrm{new}}) = (1 - \pi_{\mathrm{old}}(a_t|s_t)) \log \frac{1 - \pi_{\mathrm{old}}(a_t|s_t)}{1 - \pi_{\mathrm{old}}(a_t|s_t) l_{s_t,a_t}^\delta} - \pi_{\mathrm{old}}(a_t|s_t) \log l_{s_t,a_t}^\delta = \delta \quad (34)$$

Similarly, if $A_t > 0$, we also have $D_{\mathrm{KL}}^{s_t}(\pi_{\mathrm{old}}, \pi_{\mathrm{new}}) = \delta$.

$\square$

**Theorem 4.** *Assume that* $\max_t D_{\mathrm{KL}}^{s_t}(\pi_{\mathrm{old}}, \pi_{\mathrm{new}}^{\mathrm{PPO}}) < +\infty$ *for all t. If TRGPPO and PPO have the same hyperparameter $\epsilon$, we have:*

*(i) $u_{s_t,a_t}^\delta \geq 1 + \epsilon$ and $l_{s_t,a_t}^\delta \leq 1 - \epsilon$ for all $(s_t, a_t)$;*

*(ii) $\max_t D_{\mathrm{KL}}^{s_t}(\pi_{\mathrm{old}}, \pi_{\mathrm{new}}^{\mathrm{TRGPPO}}) = \max_t D_{\mathrm{KL}}^{s_t}(\pi_{\mathrm{old}}, \pi_{\mathrm{new}}^{\mathrm{PPO}})$;*

*(iii) $\hat{M}_{\pi_{\mathrm{old}}}(\pi_{\mathrm{new}}^{\mathrm{TRGPPO}}) \geq \hat{M}_{\pi_{\mathrm{old}}}(\pi_{\mathrm{new}}^{\mathrm{PPO}})$. Particularly, if there exists at least one $(s_t, a_t)$ such that $\pi_{\mathrm{old}}(a_t|s_t) \neq \max_{\hat{t}:A_{\hat{t}}<0} \pi_{\mathrm{old}}(a_{\hat{t}}|s_{\hat{t}})$ and $\pi_{\mathrm{old}}(a_t|s_t) \neq \max_{\hat{t}:A_{\hat{t}}>0} \pi_{\mathrm{old}}(a_{\hat{t}}|s_{\hat{t}})$, then $\hat{M}_{\pi_{\mathrm{old}}}(\pi_{\mathrm{new}}^{\mathrm{TRGPPO}}) > \hat{M}_{\pi_{\mathrm{old}}}(\pi_{\mathrm{new}}^{\mathrm{PPO}})$.*

**Proof:**

We first prove (ii). By Equation (13) and Lemma 6, we have $\max_t D_{\mathrm{KL}}^{s_t}(\pi_{\mathrm{old}}, \pi_{\mathrm{new}}^{\mathrm{TRGPPO}}) = \delta = \max_t D_{\mathrm{KL}}^{s_t}(\pi_{\mathrm{old}}, \pi_{\mathrm{new}}^{\mathrm{PPO}})$.

We then prove (i). By (ii) we have $D_{\mathrm{KL}}^{s_t}(\pi_{\mathrm{old}}, \pi_{\mathrm{new}}^{\mathrm{PPO}}) \leq \max_{\hat{t}} D_{\mathrm{KL}}^{s_{\hat{t}}}(\pi_{\mathrm{old}}, \pi_{\mathrm{new}}^{\mathrm{PPO}}) = \delta = D_{\mathrm{KL}}^{s_t}(\pi_{\mathrm{old}}, \pi_{\mathrm{new}}^{\mathrm{TRGPPO}})$ for all $(s_t, a_t)$. Thus, we have $D_{\mathrm{KL}}^{s_t}(\pi_{\mathrm{old}}, \pi_{\mathrm{new}}^{\mathrm{PPO}}) \leq D_{\mathrm{KL}}^{s_t}(\pi_{\mathrm{old}}, \pi_{\mathrm{new}}^{\mathrm{TRGPPO}})$. Indeed, by Lemma 5, we have $dl_{s_t,a_t}/dD_{\mathrm{KL}}^{s_t}(\pi_{\mathrm{old}}, \pi_{\mathrm{new}}) < 0$, $du_{s_t,a_t}/dD_{\mathrm{KL}}^{s_t}(\pi_{\mathrm{old}}, \pi_{\mathrm{new}}) > 0$. Thus, we obtain $l_{s_t,a_t}^\delta \leq 1 - \epsilon$ and $u_{s_t,a_t}^\delta \geq 1 + \epsilon$.

Particularly, by Lemma 6, if a given $(s_t, a_t)$ satisfies $\pi_{\mathrm{old}}(a_t|s_t) \neq \max_{\hat{t}:A_{\hat{t}}<0} \pi_{\mathrm{old}}(a_{\hat{t}}|s_{\hat{t}})$ and $\pi_{\mathrm{old}}(a_t|s_t) \neq \max_{\hat{t}:A_{\hat{t}}>0} \pi_{\mathrm{old}}(a_{\hat{t}}|s_{\hat{t}})$, then $D_{\mathrm{KL}}^{s_t}(\pi_{\mathrm{old}}, \pi_{\mathrm{new}}^{\mathrm{PPO}}) < D_{\mathrm{KL}}^{s_t}(\pi_{\mathrm{old}}, \pi_{\mathrm{new}}^{\mathrm{TRGPPO}})$. Hence, we have $l_{s_t,a_t}^\delta < 1 - \epsilon$ and $u_{s_t,a_t}^\delta > 1 + \epsilon$.

We finally prove (iii). By eq. (25) and eq. (27), we can get that

$$r_{\pi_{\mathrm{new}}}(s_t, a_t) A_t = \begin{cases} l_{s_t,a_t} A_t & A_t < 0 \\ \min(u_{s_t,a_t}, 1) A_t & A_t > 0 \end{cases} \quad (35)$$

By (i) we have $r_{\pi_{\mathrm{new}}^{\mathrm{TRGPPO}}}(s_t, a_t) A_t \geq r_{\pi_{\mathrm{new}}^{\mathrm{PPO}}}(s_t, a_t) A_t$ on all $s_t, a_t$. Thus, $\frac{1}{T}\sum_{t=1}^{T}\left[r_{\pi_{\mathrm{new}}^{\mathrm{TRGPPO}}}(s_t, a_t) A_t\right] \geq \frac{1}{T}\sum_{t=1}^{T}\left[r_{\pi_{\mathrm{new}}^{\mathrm{PPO}}}(s_t, a_t) A_t\right]$. By (ii) and the definition of $\hat{M}_{\pi_{\mathrm{old}}}$, we obtain $\hat{M}_{\pi_{\mathrm{old}}}(\pi_{\mathrm{new}}^{\mathrm{TRGPPO}}) \geq \hat{M}_{\pi_{\mathrm{old}}}(\pi_{\mathrm{new}}^{\mathrm{PPO}})$.

Particularly, if there exists one $(s_t, a_t)$ that satisfies $\pi_{\mathrm{old}}(a_t|s_t) \neq \max_{\hat{t}:A_{\hat{t}}<0} \pi_{\mathrm{old}}(a_{\hat{t}}|s_{\hat{t}})$ and $\pi_{\mathrm{old}}(a_t|s_t) \neq \max_{\hat{t}:A_{\hat{t}}>0} \pi_{\mathrm{old}}(a_{\hat{t}}|s_{\hat{t}})$, then we have $r_{\pi_{\mathrm{new}}^{\mathrm{TRGPPO}}}(s_t, a_t) A_t > r_{\pi_{\mathrm{new}}^{\mathrm{PPO}}}(s_t, a_t) A_t$. Hence, we obtain $\hat{M}_{\pi_{\mathrm{old}}}(\pi_{\mathrm{new}}^{\mathrm{TRGPPO}}) > \hat{M}_{\pi_{\mathrm{old}}}(\pi_{\mathrm{new}}^{\mathrm{PPO}})$.

$\square$