[Reviews · NeurIPS 2019]

Reviewer 1



The paper demonstrates a shortcmoing in PPO algorithm where a suboptimal policy that prefers suboptimal actions can diverge over time due to the constant clipping mechanism. Instead, the paper proposes an adaptive clipping mechansim that selects clipping range based on current policy, giving more chance of exploration to actions that are not preferred by the current policy. The paper demonstrates that the proposed method achieves better objective lower bound than PPO while maintaining the same KL divergence between policies. Experimental results compares the proposed method to several baselines in different settings. The paper provides elegant treatment of the problem at hand and has the potential to guide future research. The results are promising but are not very impressive unless the training time is included. I have a number of comments though. My important concerns are regarding evaluation: - Is there a reason that you are choosing average top 10 rewards? It seems more natural to use average rewards over all episodes. A related question is why you are choosing only 7 random seeds for Figure 3? - Please compare with adaptive regularization/constraint baselines such as the adaptive KL regularization version of PPO[20]. - Please add more details (perhaps in the supplementary) about your policy class as well as baseline hyper-parameters to make the results reproducible. Other comments: - Introduction: Lines 23-24 are not very accurate since there is a version of PPO that uses KL penalty without clipping[20]. - L113: Lemma 2 is not clear to me. Are you sure the LHS is not E_{\pi_{t+1}}[\pi_{t+1}(a) | \pi_0] - Algorithm 1: It seems that t <- t+1 should be in the end. - L166: while gets --> while getting. - L235: How can we guarantee that \pi^{new} exists given that it has to satisfy multiple constraints? ======================================================== I thank the authors for their response. Based on the rebuttal, I have changed my score to 7.

Reviewer 2



The paper proposes to adapt the clipping procedure of Proximal Policy Optimization (PPO) such that the lower and upper bounds are no longer constant for all states. The authors show that constant bounds cause convergence to suboptimal policies if the initial policy is initialized poorly (e.g. the probability of choosing optimal actions is small). As an alternative, the authors propose to compute state-action specific lower and upper bounds that are inside the trust region with respect to the previous policy. If the previous policy assigns a small probability to a given action, the lower and upper bounds do not need to be very tight, allowing for less agressive clipping. The adapted version of PPO, which the authors call TRGPPO, has provably better performance bounds than PPO and is validated empirically in several experiments. Since PPO is frequently used in deep reinforcement learning, correcting the suboptimal behavior of PPO seems like a relevant contribution. I think the paper properly motivates the theory, and the derivations appear correct, with the exception of one derivation that I do not follow (see below). The empirical validation is mostly expected, but there are cases for which the approximate version of PPO outperforms TRGPPO, which is a bit puzzling. Page 3: "We now give a formal illustration": This sentence appears misplaced, especially since it is immediately followed by Algorithm 1. In the supplementary material, Equation (4) correctly states the equations on the Lagrange multipliers, but I fail to see how these equations are transformed into Equation (5). "Table 1(b) lists the averaged top 10 episode rewards": this is a very crude presentation of the empirical results that does not account for variation in performance during learning. A given algorithm may have a performance peak early during learning and then suffer from catastrophic forgetting. I would much prefer to see the learning curves of the different algorithms, to help infer whether learning was stable over time. Do you have any ideas wht TRGPPO performs worse than PPO in Humanoid? This seems to contradict your theoretical results. POST-REBUTTAL: The rebuttal mostly confirmed my initial understanding: the proposed version of PPO is guaranteed to improve over the original PPO algorithm, but still does not come with any convergence guarantees, not even to an approximately optimal solution. I appreciate the effort to explain how Eq (4) is transformed into Eq (5).

Reviewer 3



Overall I think this paper presents an interesting idea in improving exploration and stability of PPO. The idea is very well presented and authors include both rigorous theoretical analysis and rich empirical experiments. Pro: 1. The idea for this paper is really well presented. The structure of the paper is well organized and the authors include simple examples and illustrations to help make the argument easily understandable. Starting by analyzing the shortcomings of PPO, the authors naturally introduce the improvement and thus make the paper easy to read. 2. The authors provide rigorous theoretical justification for the shortcomings of PPO and how the improvements in TRGPPO fix the problem. The authors also include intuitive explanation for all the theoretical results which makes the results easy to understand, 3. This paper also includes rich empirical evidence for the proposed algorithm. Besides reward performance, the authors also present analysis for some important metrics of the algorithm, which agree with the theoretical results. Con: 1. Some mathematical formulae in the paper could be better formatted. Particularly, the expressions in theorem 1 and theorem 2 could be aligned. 2. While the paper includes comparison to PPO with a single large clipping rate in section 6, it would be interesting to compare to the performance of PPO with an optimal constant clipping rate. The idea in this paper is well presented and thoroughly investigated. Overall I think the idea is novel and the contribution is significant. Despite minor flaws, I recommend publication of this paper.

[Author Response · NeurIPS 2019]

We thank anonymous reviewers for their valuable comments and suggestions.

**Comment 1: Training time (Reviewer #1)**
**Response:** We have included the training time in the paper (L318) that "Within one million timesteps, the training
wall-clock time for our TRGPPO is 33 min; for PPO, 32 min". We use several techniques to allow efficient optimization,
including the problem reduction, the DNN-approximation and problem discretization (described in Sec 5.1).

**Comment 2: Concerns of the experiment evaluation, random number and baseline (Reviewer #1 & #2 & #3)**
**Response:** We used the averaged top 10 reward following the setting in [24], which could somewhat reflect the
algorithm's ability on searching good solution but we agree it's somehow unreliable. However, we have also plotted
the learning curves in Fig. 3, which could help infer the stability of the algorithm. Per your suggestion, we have
made several revisions, listed as follows: 1) *report the averaged reward over all episodes of training*; 2) *compare with*
*the baseline of adaptive KL regularization of PPO* (and clarify the related description in the introduction); 3) *run a*
*hyperparameter sweep for $\epsilon$ of PPO over [0.1,0.6] with step 0.05*; 4) *increase the number of random seeds to 10*. We
normalized the scores for each environment so that the random policy gave a score of 0 and the best score was set to 1.
The averaged normalized scores (over 60 runs with all episodes of training for each algorithm, on 6 environments) are
as follows: ***TRGPPO: 0.629***; *PPO($\epsilon = 0.2$,default)*: 0.441; *PPO($\epsilon = 0.25$, optimal PPO)*: 0.484; *PPO-adaptiveKL*:
0.422. We will add more details of the results in the final version.

**Comment 3: Concerns about Lemma 2 (L113) and several typos (Reviewer #1)**
**Response:** Thanks for your careful reading. The correct form of the LHS of the equation in Lemma 2 should be
$\mathbb{E}_{\pi_{t+1}}[\pi_{t+1}(a)|\pi_0]$. This typo does not affect the correctness of the lemma and the remaining theoretical results in the
manuscript. We will rectify all the typos in the final version.

**Comment 4: The existence of $\pi_{\text{new}}^{\text{PPO}}$ (L235) (Reviewer #1)**
**Response:** The problem is that how we can find $\pi_{\text{new}}^{\text{PPO}} \in \Pi_{\text{new}}^{\text{PPO}}$ that achieves minimum KL divergence on all
states $s_t$, which can be formalized as $\min_{\pi \in \Pi_{\text{new}}^{\text{PPO}}} (D_{\text{KL}}^{s_1}(\pi_{\text{old}}, \pi), \ldots, D_{\text{KL}}^{s_T}(\pi_{\text{old}}, \pi))$. Note that $\pi(\cdot|s_t)$ is a conditional
probability and theoretically the optimal solution on different states are independent from each other. Thus the problem
can be optimized by independently solving $\min_{\pi(\cdot|s_t) \in \{\pi(\cdot|s_t):\pi \in \Pi_{\text{new}}^{\text{PPO}}\}} D_{\text{KL}}(\pi_{\text{old}}(\cdot|s_t), \pi(\cdot|s_t))$ for each $s_t$. The final
$\pi_{\text{new}}^{\text{PPO}}$ is obtained by integrating these independent optimal solutions $\pi_{\text{new}}^{\text{PPO}}(\cdot|s_t)$ on different state $s_t$. We have provided
detail in Appendix D and we will add more explanation in the final version.

**Comment 5: Concerns about the details of and the reproducibility of the experiment (Reviewer #1)**
**Response:** We used Gaussian and Gibbs policy for continuous and discrete tasks respectively, parametrized by a DNN.
For baseline, we used the setting recommended in the original paper. We have also submitted a link of the source code
as supplementary (L48 in the paper). We will add more details and release our code in the final version.

**Comment 6: How is Eq. (4) transformed into Eq. (5) in supplementary? (Reviewer #2)**
**Response:** To be brief, *let's number the equations in Eq. (4) by (a)-(d)*. First, by (a)(b), we have $\lambda \neq 0$, since if
$\lambda = 0$ then $\nu = 0$ (by (a)), which contradicts (b). Second, by (c) and $\lambda \neq 0$, we have $\sum_{a \in \mathcal{A}} p'_a \log(p'_a/p_a) = \delta$.
Third, taking (a) into (d), we have $p'_a/p_a = \nu/\lambda = (1 - p'_{a_t})/(1 - p_{a_t})$ for $a \neq a_t$. Then, taking this equation into
$\sum_{a \in \mathcal{A}} p'_a \log(p'_a/p_a) = \delta$, we obtain Eq. (5). We will add more details in the final version.

**Comment 7: The performance on Humanoid (Reviewer #2)**
**Response:** One possible explanation is that the larger clipping range of our TRGPPO may make it suffer from the
noisy estimated advantage values, especially at the later training phase where the advantage values are large and noisy.
This issue could be addressed using our adaptive clipping scheme by taking the trade-off between exploration and
stability into account. In particular, in the revised version, we have implemented two variants of TRGPPO: linearly
decaying $\epsilon$ from 0.2 to 0.1 (named by *TRGPPO-decay*) or clipping the clipping ranges (named by *TRGPPO-clipping*),
i.e., $l_{s,a}^{\delta,\epsilon,\epsilon_t} = clip(l_{s,a}^\delta, \epsilon_t, \epsilon)$, $u_{s,a}^{\delta,\epsilon,\epsilon_t} = clip(l_{s,a}^\delta, 1/\epsilon, 1/\epsilon_t)$, where $0 < \epsilon < 1$ and $\epsilon_t = \epsilon t/T$ are parameters to control
the level of the clipping ranges, $t$ and $T$ are the current and total training iterations respectively. Both these two
methods could improve the reward and sample efficiency. The averaged episode rewards over all episodes of training on
Humanoid are as follows: *TRGPPO-decay*: 3013.3; ***TRGPPO-clipping: 3148.1***; *PPO*: 2944.2. The timesteps ($\times 10^3$)
to hit the threshold are as follows: *TRGPPO-decay*: 7514; ***TRGPPO-clipping: 7132***; *PPO*: 9088.

**Comment 8: Prove that TRGPPO converges to the optimal policy (Reviewer #2)**
**Response:** Thanks for your suggestion. It's interesting to prove such convergence property.However, there seems
does not exist closed-form of our clipping range in Eq. (5), making it hard to measure the improvement of $\Delta_{\pi_0,t}^{\text{TRGPPO}}$
by the explicit form of $\mathbb{E}_{\pi_{t+1}}[\pi_{t+1}(a_{\text{opt}})|\pi_t]$ (see Eq. (3)). Alternatively, we plan to work on this by analyzing the
corresponding bound for each term in Eq.(3).

**Comment 9: Some mathematical formulae in the paper could be better formatted. (Reviewer #3)**
**Response:** Thanks for your comment. We will polish mathematic notions and align expressions in the final version.

[Meta-Review · NeurIPS 2019]

This paper provides an analysis on the exploration behavior of PPO, and shows that show that PPO is prone to suffer from the risk of lack of exploration. Specifically, the paper shows that Proximal Policy Optimization (PPO) converges to a suboptimal policy if the policy initialization is not done correctly. The authors solve this issues by proposing an adapted version of the clipping mechanism for PPO. The paper contains both a theoretical analysis of the new exploration technique and an empirical analysis that clearly demonstrates the advantages of the proposed method over PPO. The authors should have compared to other existing RL algorithms and not just PPO.